# Structural insight into the allosteric inhibition of human sodium-calcium exchanger NCX1 by XIP and SEA0400

Yanli Dong [1,6], Zhuoya Yu[1,2,3,6], Yue Li [1,2,3,6], Bo Huang[4], Qinru Bai[1,2,3], Yiwei Gao [1,2,3], Qihao Chen[1,2,3], Na Li[5], Lingli He[1] & Yan Zhao [1,2,3✉]

## Abstract

Sodium-calcium exchanger proteins influence calcium homeostasis in many cell types and participate in a wide range of physiological and pathological processes. Here, we elucidate the cryo-EM structure of the human $Na^+/Ca^{2+}$ exchanger NCX1.3 in the presence of a specific inhibitor, SEA0400. Conserved ion-coordinating residues are exposed on the cytoplasmic face of NCX1.3, indicating that the observed structure is stabilized in an inward-facing conformation. We show how regulatory calcium-binding domains (CBDs) assemble with the ion-translocation transmembrane domain (TMD). The exchanger-inhibitory peptide (XIP) is trapped within a groove between the TMD and CBD2 and predicted to clash with gating helices TMs[1/6] at the outward-facing state, thus hindering conformational transition and promoting inactivation of the transporter. A bound SEA0400 molecule stiffens helix TM2ab and affects conformational rearrangements of TM2ab that are associated with the ion-exchange reaction, thus allosterically attenuating $Ca^{2+}$-uptake activity of NCX1.3.

**Keywords** Sodium-calcium Exchanger; Calcium Homeostasis; Allosteric Inhibition; SEA0400; Exchanger Inhibitory Peptide (XIP)
**Subject Category** Structural Biology

## Introduction

The $Na^+/Ca^{2+}$ exchanger (NCX), belonging to the CaCA ($Ca^{2+}$/ Cation antiporter) superfamily of secondary active transporters, forms one of major pathways to mediate $Ca^{2+}$ fluxes across the cell membrane and significantly contributes to the regulation of $Ca^{2+}$-dependent events in various cell types (Blaustein and Lederer, 1999; Hilgemann et al, 1991). The NCX is a bi-directional and reversible transporter, and utilizes the electrochemical gradient of $Na^+$ to exchange 3 $Na^+$ ions for 1 $Ca^{2+}$ across the membrane, thus performing electrogenic transport (Blaustein and Russell, 1975; Crespo et al, 1990; Hilgemann et al, 1991; Kang and Hilgemann, 2004; Lagnado and McNaughton, 1990; Reeves and Hale, 1984). Depending on prevailing gradient of $Na^+$ and membrane potential, the NCX operates in either the forward mode, coupling the uphill extrusion of $Ca^{2+}$ to the downhill influx of $Na^+$ ions, or in the reverse mode, coupling the extrusion of $Na^+$ to the influx of $Ca^{2+}$ ions, both of which assume important physiological functions. In mammalian, three NCX genes (*SLC8A1, SLC8A2,* and *SLC8A3*), encoding NCX1–3, have been identified and cloned (Li et al, 1994; Nicoll et al, 1996; Quednau et al, 1997). Among the three NCX isoforms, the NCX1 is the most highly characterized, expressing distinct splice variants in the heart, brain, and kidney(Khananshvili, 2014). It plays pivotal roles in the regulation of cellular $Ca^{2+}$ homeostasis to modulate many fundamental physiological events, including excitation-contraction coupling of cardiac muscle (Bers, 2002; Sipido et al, 2002), the brain's long-term potentiation and learning (Boscia et al, 2006; Molinaro et al, 2008; Pignataro et al, 2004), blood pressure regulation (Zhang et al, 2010), and neurotransmitter secretion (Jeon et al, 2003; Molinaro et al, 2008). Thus, alteration in NCX expression and regulation has been implicated in many cardiac diseases and brain disorders, such as ischemia-reperfusion injury (Chen and Li, 2012b), stroke damage (Molinaro et al, 2016; Vinciguerra et al, 2014), and neonatal hypoxia (Cerullo et al, 2018).

Crystal structures of bacterial NCX from archaebacterial *Methanococcus jannaschii* (mjNCX) reveal the arrangement of ten transmembrane helices and four ion binding sites formed by two so-called α-repeats which are accessible via two passageways (Liao et al, 2012; Liao et al, 2016). However, the cytosolic loop between TM5 and TM6 of mjNCX only contains 12 residues, and is much shorter than the corresponding regulatory f-loop of human NCX. The f-loop consists of two tandem calcium-binding domains (CBD1 and CBD2), which exert allosteric regulation of NCX activity by binding cytosolic $Ca^{2+}$. Multiple structures of isolated CBD1 and CBD2 determined by X-ray crystallography or NMR methods illustrate structural features and conformational changes

[1]Key Laboratory of Biomacromolecules (CAS), National Laboratory of Biomacromolecules, CAS Center for Excellence in Biomacromolecules, Institute of Biophysics, Chinese Academy of Sciences, Beijing 100101, China. [2]State Key Laboratory of Brain and Cognitive Science, Institute of Biophysics, Chinese Academy of Sciences, 15 Datun Road, Beijing 100101, China. [3]College of Life Sciences, University of Chinese Academy of Sciences, Beijing 100049, China. [4]Beijing StoneWise Technology Co Ltd., 15 Haidian street, Haidian district, Beijing, China. [5]Heart Center and Beijing Key Laboratory of Hypertension, Beijing Chaoyang Hospital, Capital Medical University, Beijing 100020, China. [6]These authors contributed equally: Yanli Dong, Zhuoya Yu, Yue Li. ✉E-mail: zhaoy@ibp.ac.cn

in the presence or absence of calcium ion (Besserer et al, 2007; Giladi et al, 2012; Hilge et al, 2009a; Hilge et al, 2006; Nicoll et al, 2006a). In addition to the CBD1/2, the exchanger inhibitory peptide (XIP) region located within the f-loop has also been determined to block NCX activity in response to high concentration of cytosolic sodium ion ($[Na^+]_{in}$) (Li et al, 1991; Matsuoka et al, 1997b). However, the mechanism of how these regions regulate transport activity is still unknown due to a lack of structural insight of intact mammalian NCX. Moreover, increasing evidence demonstrated that intracellular $Ca^{2+}$ overload or $Na^+$ accumulation through NCX leads to a variety of pathological conditions, and thus many potent and specific NCX inhibitors, such as benzyloxyphenyl derivatives (including SEA0400)(Matsuda et al, 2001), have been developed and showed therapeutic potential to treat related diseases. However, the structural basis of how these inhibitors bind to and inhibit the transport activity remains elusive.

In this work, we purified the recombinant human full-length wild-type (WT) NCX1.3 and determined its complex structure bound with SEA0400 by using the cryo-electron microscopy (cryo-EM) method. The structure of the NCX1.3 clearly reveals the assembly of the transmembrane domain and its regulatory cytosolic domain. The conserved residues for coordinating $Na^+$ or $Ca^{2+}$ are accessible from the intracellular side, indicating that the current NCX1.3 structure is trapped in an inward-facing conformation, which is stabilized by both the XIP region and SEA0400 antagonist. The XIP region is located at juxtamembrane region and forms extensive interactions with surrounding intracellular parts of TM helices as well as CBD2. Further electrophysiological experiments suggest that disruption of these interactions of XIP alleviates $[Na^+]_{in}$-dependent inactivation of the NCX1.3. The SEA0400 inhibitor is buried in a pocket closed to the ion passageway and allosterically blocks uptake activity of NCX1.3 by restricting local backbone dynamics of TM2ab as well as hindering conformational change of TM2ab during the transport cycle.

## Results

### Architecture of the human NCX1.3

To investigate the structural basis of ion-transport and allosteric regulation mechanisms of NCX, we fused NCX1.3 with a fluorescence protein mCherry after the N-terminus signal peptide (residues 1–35 amino acids) to monitor the expression of the NCX1.3 (Appendix Table S1). We carried out single-cell $Ca^{2+}$ imaging and whole-cell patch-clamp assays using HEK293T expressing NCX1.3. Under the reverse mode, the cells showed $Ca^{2+}$ influx, suggesting that the construct used for expression is functional (Appendix Fig. S1A,B). Subsequently, we expressed the NCX1.3 in HEK293 cells and purified it in lauryl maltose neopentyl glycol (LMNG) detergent. Size-exclusion chromatography (SEC) displayed monodisperse peaks, and SDS-PAGE result confirmed the intactness of the full-length NCX1.3 (Appendix Fig. S1C,D). However, there are several distinct, closely spaced bands around the 70 kD and 50 kD marker bands. To identify these protein bands, we excised a broader region (Appendix Fig. S1D) for the mass spectrometry analysis. The results suggest that the proteins with a lower molecular weight represent degraded NCX1.3, heat shock protein 70 (HSP70), and a few other contaminant proteins

(Appendix Fig. S1E). Meanwhile, the thermofluor assay demonstrated that the NCX1.3-selective inhibitor SEA0400 can raise the melting temperature of NCX1.3 by ~10 °C (Appendix Fig. S1F). Therefore, during protein expression, purification, and grid preparation, we included the SEA0400 to improve the stability and conformational homogeneity of the NCX1.3 sample.

We carried out single-particle cryo-EM studies and collected 12,573 movie stacks. The transmembrane-helices feature and characteristic soluble domains of NCX1.3 were clearly visualized on the 2D classes (Appendix Fig. S2A). Next, the initial 3D models were generated by *ab*-initio reconstruction without reference and used for subsequent 3D classification. The final selected particles were subjected to 3D refinement and yielded an EM map of 3.5-Å resolution, which includes many well-resolved features and clear densities of main-chains, side-chains (Fig. 1A and Appendix Figs. S2A–D and Table S2). Model building of the NCX1.3 is carried out by docking of the crystal structures of mjNCX (PBD ID: 3V5S), CBD1 (PDB ID: 2DPK), and CBD2 (PDB ID: 2KLT) structures into the map, followed by manually adjustment and real-space refinement (Appendix Fig. S2E).

Human NCX1.3 consists of 937 amino-acid residues; our NCX1.3 model is, however, composed of residues 51–283, 405–501, 518–637, 654–720, and 736–937. Several long loops are invisible presumably due to conformational heterogeneity. NCX1.3 structure consists of 10 transmembrane helices (TMs 1–10) and the large intracellular regulatory domains CBD1 and CBD2, with both N- and C-termini located at the extracellular side (Fig. 1B–D). The NCX1.3 structure is ~130 Å in height, and the transmembrane region of NCX1.3 is ~50-Å long and 40-Å wide (Fig. 1B,C). Its TM1–TM5 region is related to TM6–TM10 by a pseudo 2-fold symmetry, with the rotation axis parallel to the membrane plane, in line with previously reported prokaryotic structure of mjNCX (Liao et al, 2012; Liao et al, 2016). TM2–3 and TM7–8 regions contain the signature α1- and α2-repeats, respectively, which are conserved in the whole NCX family and responsible for ion binding and translocation (Appendix Fig. S3). They are characteristically unwound, resulting in fragments TM2ab, TM2c, TM3a, TM3b, TM7a, TM7b, TM7c, TM8a, and TM8b, and the two unwound helices cross each other in the vicinity of the pseudo 2-fold axis (Fig. 1B–D). Loops connecting TM helices at the extracellular side are shorter in general. However, there are some long loops at the intracellular side. For instance, the linker between TM1 and TM2 forms a β-sheet (β1 and β2) and protrudes into the cytosol (Fig. 1B,D). Especially, the NCX1.3 contains a large cytosolic region between the two halves of the TMD with residues ranging from 260 to 683, of which exchanger inhibitory peptide (XIP, residues 260–283), CBD1 (residues 405–536), and CBD2 (residues 537–683) were clearly resolved in our structure. The XIP following TM5 is positioned proximal to the membrane and forms antiparallel β-strands (β3 and β4) (Fig. 1D). Previous studies demonstrated that both of CBD1 and CBD2 have an immunoglobulin-like β-sandwich fold, composed of seven antiparallel β-strands with their $Ca^{2+}$-binding sites located at the ends of β-strands (Besserer et al, 2007; Giladi et al, 2012; Hilge et al, 2009a; Hilge et al, 2006; Nicoll et al, 2006a). The structure of NCX1.3 illustrates that the CBD1 extends deep into the cytosol while the CBD2 locates close to the cytoplasmic membrane surface (Fig. 1A,B). The CBD1 and CBD2 are organized in a head-to-tail tandem through a short linker, generating a ~126° angle between CBD1 and CBD2 domains

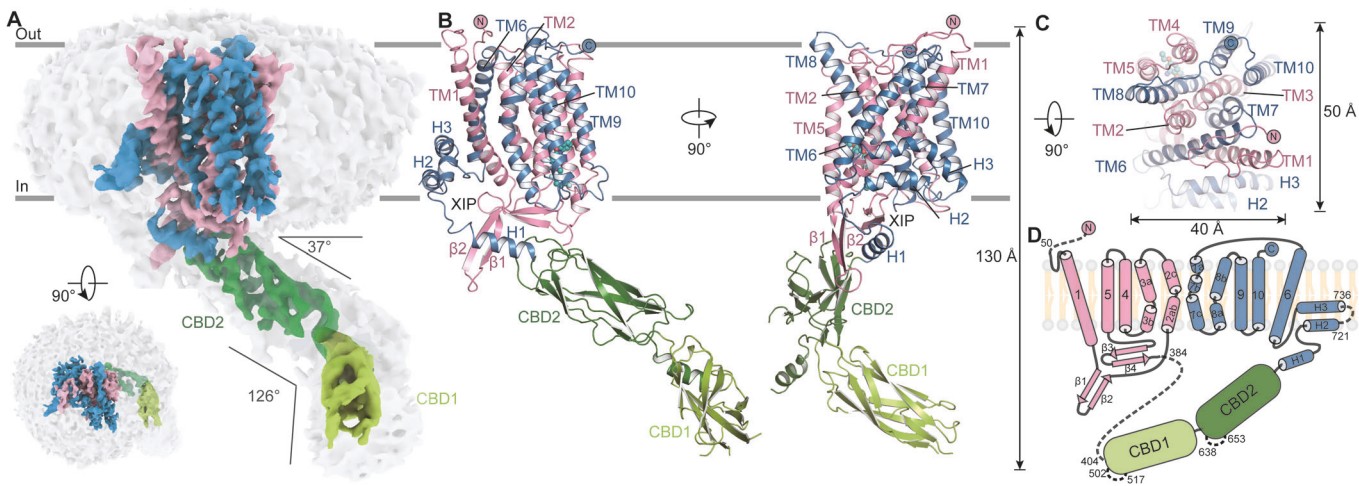

**Figure 1. Architecture of human NCX1.3 exchanger.**

(A–C) Cryo-EM map and atomic model of human NCX1.3 exchanger, viewed in parallel to the membrane plane and from the extracellular side, respectively. TMD of NCX1.3 is colored in blue and pink. CBD1 and CBD2 are colored in light and deep green, respectively. Outer surface of the map was overlaid as a gray transparent surface at a lower density threshold. Cell membrane is represented by gray lines. Height and width of the NCX1.3, and angle between the CBD2 and the membrane plane or the CBD1 domains are indicated. (D) Topology model of the NCX1.3 exchanger. The color code was the same as Fig. 1A.

(Figs. 1A and EV1). The CBD1 domain of the NCX1 protein contains four calcium ion binding sites, with $Ca^{2+}$ ions displaying a strong affinity for CBD1 ($Kd_{Ca1/Ca2}$: ~10 μM, $Kd_{Ca3/Ca4}$: ~0.4 μM) (Hilge et al, 2009a). However, due to the moderate resolution of our resolved NCX1.3 map, we were unable to observe the density corresponding to $Ca^{2+}$ ions. A previous study suggested that the upper half of isolated CBD1 loses its structural integrity in the absence of $Ca^{2+}$ (Hilge et al, 2006). In contrast, the CBD1 domain in our NCX1.3 structure exhibits a well-defined conformation, with an RMSD of ~0.9 Å for 116 Cα-pairs when compared to the isolated $Ca^{2+}$-bound CBD1 (Fig. EV1A). Considering these findings, along with the fact that we added ~1 mM $Ca^{2+}$ during the sample preparation, we speculate that $Ca^{2+}$ ions likely bind to the CBD1 domain in our elucidated NCX1.3 structure. Nevertheless, we cannot entirely rule out the possibility that $Ca^{2+}$ may remain unbound or be located in alternative positions. Further investigation is required to understand the calcium binding site and its regulatory roles. The CBD1 and CBD2 in our NCX1.3 structure exhibit a V-shaped conformation similar to the CBD1-2 structure from NCX1.4 (Giladi et al, 2012) (Fig. EV1B). By superimposing the invariant CBD1 domains of NCX1.3, NCX1.4, and NCX1.1(its structure was available online during our review), the angle between CBD1 and CBD2 decreases by 7° in NCX1.4 (exon AD), but increase by 10° in NCX1.1 (exon ACDEF), which may be attributed to the previously reported splicing region within CBD2 (Kofuji et al, 1994; Quednau et al, 1997; Xue et al, 2023) (Figs. EV1A–D). The CBD2 connects to TM6 via three helices (H1−H3), forming a ~37° angle between CBD2 and the membrane plane (Fig. 1A), which is larger than that observed in NCX1.1, possibly due to different local interactions between the TMD region and mutually exclusive exons A (NCX1.1) and B (NCX1.3), or the truncation of NCX1.1 between CBD1 and TMD (Fig. EV1D). In the 3D structure, the H1 is flanked by β1, β2, and XIP peptide. Strikingly, the H2 and H3 form a reentrant helical pair penetrating half way into the membrane, and the H3 wraps around the TM1 helix (Fig. 1B–D).

## NCX1.3 is trapped in an inward-facing conformational state

The TM2−3 and TM7−8 include 12 ion-coordinating residues that form a signature motif containing four binding sites to recognize $Na^+$ and $Ca^{2+}$ (Appendix Fig. S3). Among them, the Ala141 in TM2 ($A141^{TM2}$), $S144^{TM2}$, $D813^{TM7}$, and $S837^{TM8}$ residues are accessible from the intracellular side in our structure, whereas the remaining residues are completely buried within TMD (Fig. 2A). Thus, our structure adopts an inward-facing conformation ($NCX1.3^{IF}$). To understand the conformational change during ion exchanging, the outward-facing structure of prokaryotic mjNCX ($mjNCX^{OF}$) is overlaid onto the NCX structure using tightly packed core domain (TM2−5 and TM7−10) as a reference (Fig. 2B). In this comparison, whereas the core domains from both structures are roughly superimposable, two loosely packed helices TM1 and TM6 ($TMs^{1/6}$) undergo remarkable conformational change. In particular, and yield ~25° and ~30° counterclockwise rotation viewed from the extracellular side, respectively (Fig. 2B). In addition, the NCX exhibits inverted two-fold symmetry and the arrangement of TM helices of N- and C-terminal halves of NCX1.3 are nearly identical (Fig. EV2A). We created an outward-facing model of NCX1.3 ($NCX1.3^{OF*}$) by superimposing TM2-TM5 on TM7-TM10, and vice versa (Liao et al, 2012; Forrest et al, 2008; Hu et al, 2011) (Fig. EV2A). Comparing the $NCX1.3^{IF}$ structure with $NCX1.3^{OF*}$ using the compact core domain as a reference, the gating helices $TMs^{1/6}$ also shows dramatically sliding motion relative to the core domain (Fig. EV2B). The relative sliding movement between $TMs^{1/6}$ helices and the core domain alternatively exposes the ion binding sites to the intracellular and extracellular sides, resulting in the conformational transition between the inward- and outward-facing states

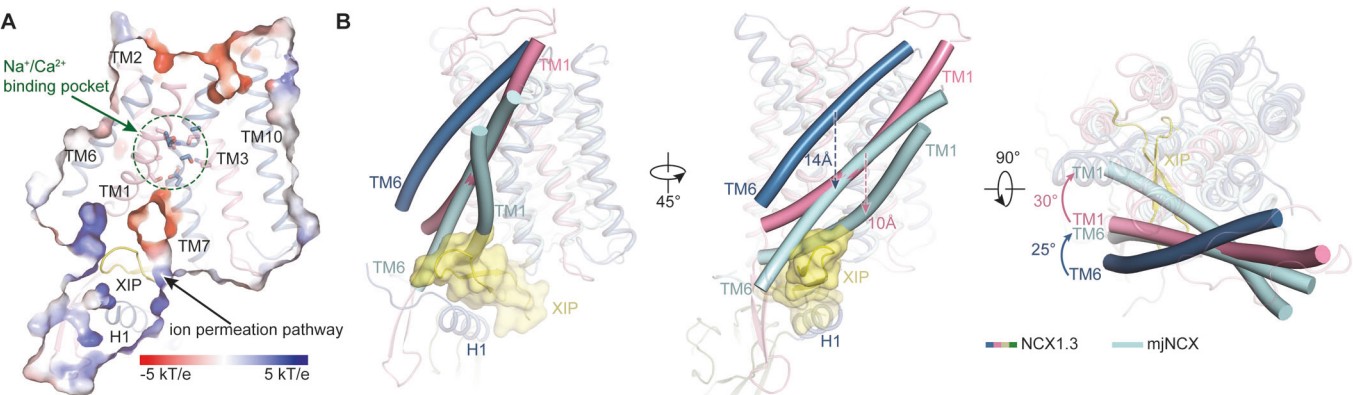

**Figure 2. Conformational changes between the inward- and outward-facing of NCX1.3.**

(A) Slice-through electrostatic surface of NCX1.3 in the inward-facing conformation. TMD and XIP region are shown in cartoon. XIP region is colored in yellow. The $Na^+$/$Ca^{2+}$ binding pocket is indicated by a green circular dotted circle. The ion permeation pathway is indicated by the black arrow, and key residues in the $Na^+$/$Ca^{2+}$ binding pocket are showed in sticks. (B) Superimposition of NCX1.3 (blue and red) and mjNCX (PDB ID: 3V5U, cyan), using the core domain (TM2−5 and TM7−10) as a reference, viewed in parallel to the membrane plane and the extracellular side, respectively. TM1 and TM6 are shown as bent cylindrical helices. Displacements and rotation of TM1 and TM6 are indicated. The ion permeation pathway at the intracellular side of NCX1.3 and the extracellular side of mjNCX are shaded using yellow and cyan triangles, respectively.

(Movie EV1), consistent with the mechanism of ion-exchange reaction proposed for prokaryotic $Ca^{2+}$/$H^+$ antiporters (Waight et al, 2013; Wu et al, 2013a).

## Inactivation regulation of NCX1.3 by the XIP region

Previous reports demonstrate that the XIP region mediates $[Na^+]_{in}$-dependent inactivation and exerts an inhibitory effect on NCX activity (He et al, 2000; Hilgemann et al, 1992; Li et al, 1991; Matsuoka et al, 1993). The XIP region, C-terminal to TM5, is clearly resolved in our structure. According to current structure, we re-designated the XIP region, which comprises 23 amino acid residues (residues 260–283) (Fig. 3A). Structural comparison between NCX1.3[IF] and mjNCX[OF] also reveals a steric hinderance between the XIP region of NCX1.3 and TMs[1/6] helices of mjNCX[OF] (Fig. 2B), suggesting that the XIP region in its current position would prohibit conformational transition from the inward-facing to outward-facing state. Given that the sample was prepared in the presence of high sodium concentration (~150 mM), we thus speculate that our current structure represents an XIP-inhibited inward-facing conformational state. Nevertheless, due to moderate resolution of the map, no bound sodium ion is confirmed in our structure. The XIP region is rich of positively charged residues and folds into two antiparallel β-strands (Fig. 3A–C). It inserts into a juxtamembrane, highly negatively charged groove formed by TM1, β2, TM2, TM7, TM8, H1, and CBD2, and is largely stabilized by complementary electrostatic interactions (Fig. 3C). In particular, three hydrogen bonds are identified to knit the antiparallel XIP and β2[TM1-2] together (Fig. EV3A,B). The β2[TM1-2] functions as an arm to hold the XIP region close to the cytosol membrane surface (Fig. EV3A,B). The side chain of K264[XIP] protrudes into a negatively charged pocket, which is contributed by acidic residues from CBD2, including E551, D612, E614, and E615 (Fig. 3C). A previous report indicates that the point mutation of K229Q of canine NCX1.1 abolishes $[Na^+]_{in}$-dependent inactivation of the NCX, which

corresponds to K264 in human NCX1.3 (Matsuoka et al, 1997a). The position of XIP inside its binding groove is also stabilized by multiple pairs of hydrogen bonding interactions, namely the R265[XIP]-E685[H1], R267[XIP]-E132[β2-TM2], Q271[XIP]-E104[TM1], and D281[XIP]-Q825[TM7-8] pairs (Fig. 3C). Furthermore, as the side chains of I275[XIP] and I276[XIP] face two sides of the β-strand, I275 is cradled within a hydrophobic pocket created by the side-chains of F686[H1], V690[H1], L693[H1], V127[β2], and I129[β2-TM2], and I276 interacts with W130[β2-TM2] (Figs. 3B,C and EV3A,B). To further verify the functional roles of the XIP region, we performed whole-cell clamp patch experiments on NCX1.3 variants of mutations in this region. The experiments were carried out in the reverse mode using an internal solution containing high concentration $Na^+$ (120 mM). The WT NCX1.3 shows that its transport-associated current is profoundly decayed with negligible steady-state transport activity (Fig. 3D and Appendix Fig. S1A). We employed R10 value to denote the inactivation of current, which is the ratio of the mean current density at the end of the 10-second test pulse to the peak amplitude. The R10 value is 0.09 ± 0.03 for WT NCX1.3 (Fig. 3E). In contrast, after deletion of the XIP region, the steady state current of the NCX1.3[ΔXIP] mutant is remarkably increased with an R10 value of 0.4 ± 0.03 (Fig. 3D,E). This result is consistent with previous reports suggesting that the XIP is critical for the inactivation of the NCX1.3 (Chin et al, 1993; Li et al, 1991; Matsuoka et al, 1997b). We also deleted the residues 110–128 on β1 and β2 between TM1 and TM2 helices (NCX1.3[Δβ1β2]) or substitute the I129[β2] and W130[β2] with proline (NCX1.3[IWPP]) to disrupt interactions between β2 and XIP. It turns out that these two mutants failed to uptake $Ca^{2+}$ into cells, suggesting that the β1 and β2 connecting TM1 and TM2 is important for transport activity of the NCX1.3 (Fig. EV3C,D). Subsequently, we designed a double mutant to disrupt stabilizing-interactions, replacing the hydrophobic residues I275[XIP] and I276[XIP] with hydrophilic glutamine (NCX1.3[IIQQ]). It turns out that the NCX1.3[IIQQ] variant maintains a significant steady state current (~28% of the peak current) (Fig. 3D,E), indicating that the above

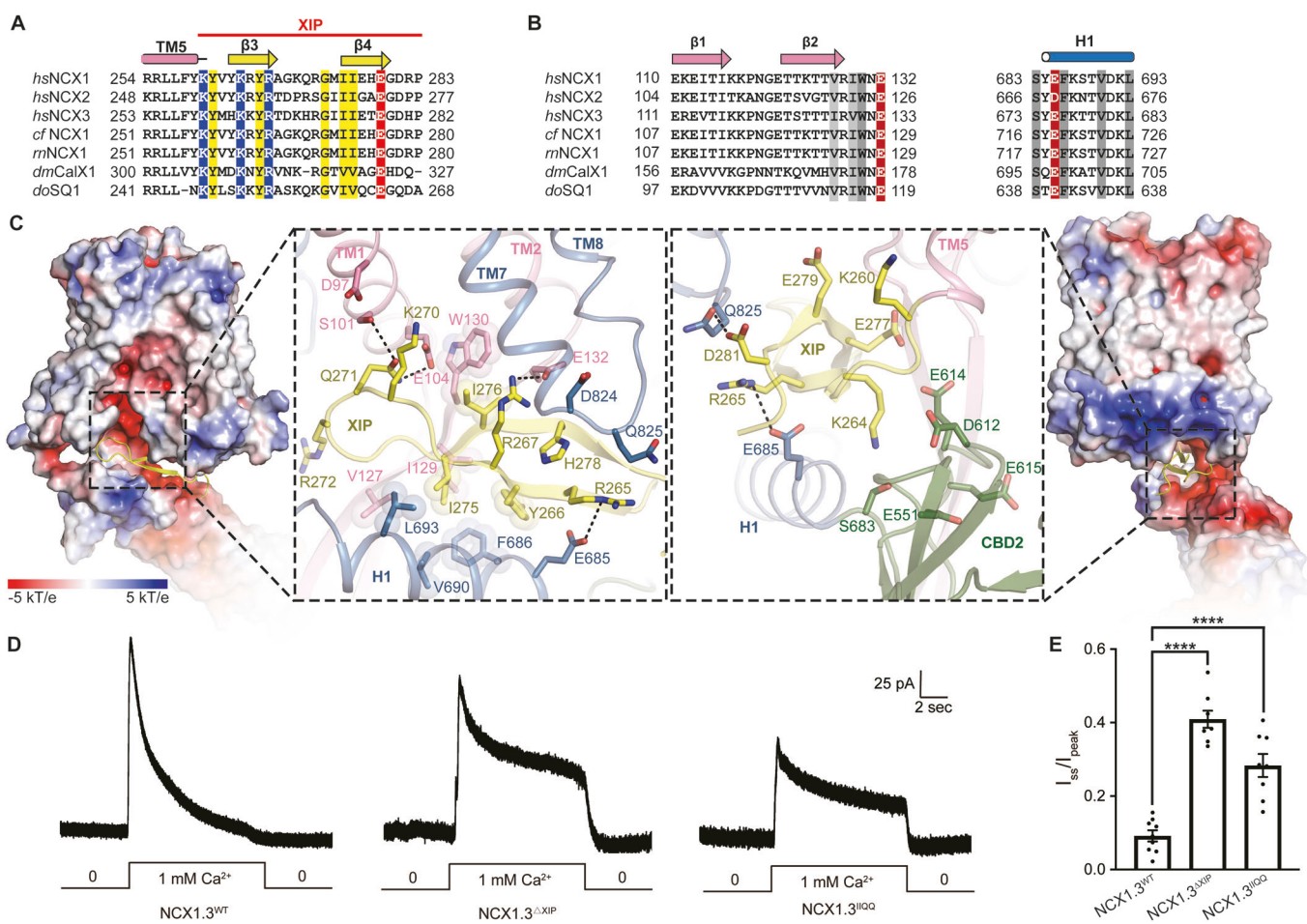

**Figure 3. Regulation of NCX1.3 by XIP region.**

(A) Sequence alignment of the XIP region among NCX homologues. Secondary structures of hsNCX1 are marked above the sequence alignment. The sequence of redefined XIP region is delineated with a red line above the secondary structure markers. Conserved residues in the XIP region are highlighted using yellow, blue (positively charged) or red (negatively charged). Sequence alignment of β1, β2, and H1 region among NCX homologues. Conserved residues in close contact with XIP are highlighted using gray or red (negatively charged). (B) Electrostatic potential surface around XIP and zoomed-in interactions between XIP and surrounding residues, viewed from two opposite directions. Key residues are shown as sticks. (C) Representative outward currents of NCX1.3$^{WT}$, NCX1.3$^{XIP}$, and NCX1.3$^{IIQQ}$, elicited by changing the extracellular Ca$^{2+}$ from 0 to 1 mM for 10 s. (D) The R10 values of the ratio of steady state to outward peak currents. NCX1.3$^{WT}$, $n = 9$; NCX1.3$^{XIP}$, $n = 8$; and NCX1.3$^{IIQQ}$, $n = 8$. Error bars indicate SEM and a black dot represent one patch record. ****$P < 0.0001$ vs. NCX1.3$^{WT}$, by two-tailed Student's t-test. (E) The R10 values of the ratio of steady state to outward peak currents. NCX1.3$^{WT}$, $n = 9$; NCX1.3$^{XIP}$, $n = 8$; and NCX1.3$^{IIQQ}$, $n = 8$. Error bars indicate SEM and a black dot represent one patch record. ****$P < 0.0001$ vs. NCX1.3$^{WT}$, by two-tailed Student's t-test. Source data are available online for this figure.

mentioned hydrophobic interactions of the XIP with H1 and β2 loop are critical for [Na$^+$]$_{in}$-dependent inactivation of the NCX1.3.

## Antagonism of NCX1.3 by SEA0400

The benzyloxyphenyl derivative SEA0400 preferentially inhibits Ca$^{2+}$ uptake by the reverse mode of NCX transporters (Matsuda et al, 2001). It is composed of ethoxyaniline, methxyphenol, and difluorophenyl groups (Fig. 4A). In the current structure, an extra density is clearly determined, which is well fitted with a SEA0400 molecule (Fig. 4B). The SEA0400 molecule is allocated within a negatively charged cavity surrounded by TM2ab, TM4, TM5, and TM8 (Fig. 4B,C). Strikingly, the SEA0400 binding pocket is not overlapped with the proposed ion passageway from the cytosol to the substrate-binding sites inside TMD (Fig. 4C). In fact, the pocket

and passageway are separated by the TM2ab helix, indicating that SEA0400 inhibits transport activity through an allosteric fashion (Fig. 4C). The ethoxyaniline group of SEA0400 forms hydrogen bonding interaction with T138$^{TM2ab}$ and hydrophobic interactions with F248$^{TM5}$. The methxyphenol group is positioned proximal to G832$^{TM8}$ and G836$^{TM8}$. Difluorophenyl forms hydrophobic interactions with L134$^{TM2ab}$, L137$^{TM2ab}$, and V203$^{TM4}$ (Fig. 3D).

Considering the elongated, strip-shaped structure of SEA0400 and moderate map quality for this molecule, we carried out molecular dynamics simulations to validate its binding pose. These simulations were initially performed using the cryo-EM structure of SEA0400-bound NCX1.3 (NCX1.3$^{EM}$). In addition, we also studied an alternative model in which SEA0400 binds to the same pocket but adopts an inverted orientation compared to what was observed in the cryo-EM structure (NCX1.3$^{rev}$) (Appendix Fig. S4A). Three independent

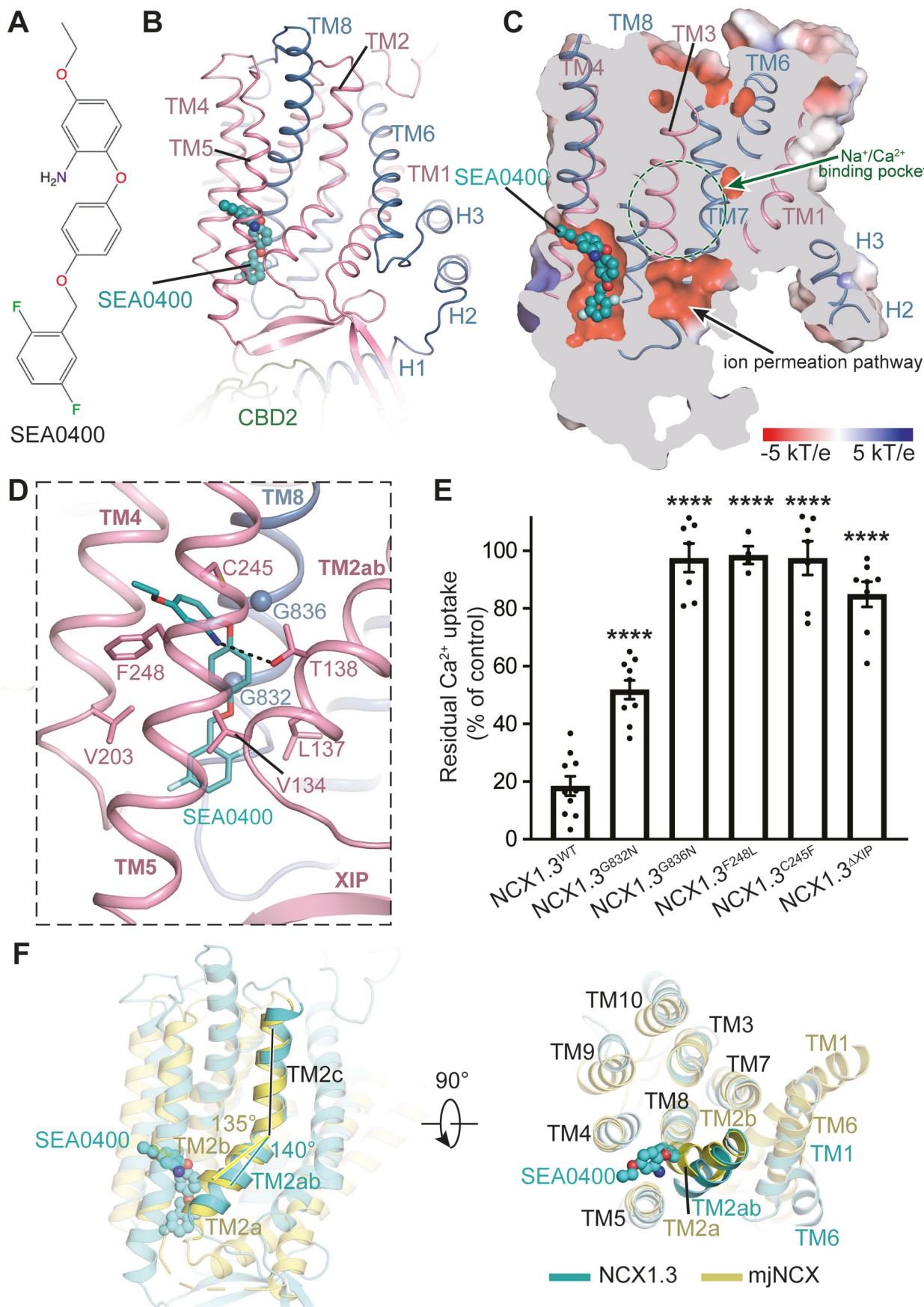

**Figure 4. Allosteric inhibition of NCX1.3 by SEA0400.**

(A) Chemical structure of SEA0400. (B) SEA0400 in the structure of NCX1.3, shown as sticks and overlaid using its EM density. TMD and XIP region are shown in cartoon. (C) Slice-through electrostatic surface of NCX1.3. SEA0400 binds at a negative potential cavity and is shown as spheres. The $Na^+/Ca^{2+}$ binding pocket is indicated by a green dashed circle. (D) The binding site of SEA0400. Residues in close contact with SEA0400 are displayed in sticks. Black dashed lines represent hydrogen bonds. G832 and G836 are shown as spheres. (E) Residual $Ca^{2+}$ uptake (% of control) after applying 10 μM SEA0400. Values were determined by dividing the change in Fura-2 ratio (peak minus baseline) after the application of 10 μM SEA0400 by the change in Fura-2 ratio before SEA0400 treatment. $NCX1.3^{WT}$, $n = 10$; $NCX1.3^{C245F}$, $n = 7$, $NCX1.3^{F248L}$, $n = 4$; $NCX1.3^{G832N}$, $n = 10$; $NCX1.3^{G836N}$, $n = 7$; and $NCX1.3^{\Delta XIP}$, $n = 8$. Error bars are defined as mean ± SEM from at least three independent experiments. ****$P < 0.0001$ vs. $NCX1.3^{WT}$, by two-tailed Student's $t$-test. (F) Structural comparison of the TM2 between NCX1.3 (in rainbow) and mjNCX (in cyan), using core domain (TM2−5 and TM7−10) as a reference. Bending angle of the TM2 (TM2ab vs. TM2c in NCX1.3, TM2a/2b vs. TM2c in mjNCX) are indicated. SEA0400 is shown in spheres. Source data are available online for this figure.

trajectories, each lasting 100 ns, were conducted for both $NCX1.3^{EM}$ and $NCX1.3^{rev}$, respectively. Subsequently, the trajectories for each group were combined and subjected to a clustering analysis (Appendix Fig S4B and C). The analysis focused on the ligand and protein residues within a 5 Å radius of the ligand, using a 1.5 Å R.M.S.D. cut-off. The trajectory clustering for $NCX1.3^{EM}$ yielded only two clusters, with cluster 1 being the predominant conformation. This conformation is close to the pose of SEA0400 observed in our cryo-EM structure, indicating the high stability of SEA0400 binding in the observed cryo-EM pose (Appendix. Fig. S4C,D). In contrast, the trajectory for $NCX1.3^{rev}$ produced eight distinct clusters, and the pose of SEA0400 exhibited significant deviations among these clusters, suggesting the low stability of such a binding pose (Appendix. Fig. S4C,D). These findings provide support for the binding pose of SEA0400 in the cryo-EM structure.

To verify the binding pocket of SEA0400, we designed several mutants to disturb SEA0400 binding and carried out calcium imaging experiment using Fura-2 as an indicator to monitor changes of $Na^+$-dependent $Ca^{2+}$ uptake between the wild-type and mutant variants. In the presence of extracellular $Ca^{2+}$, we found that confluent HEK293T cells expressing $NCX1.3^{WT}$ show a robust $Ca^{2+}$ increase when 140-mM extracellular $Na^+$ is replaced with equimolar N-ethyldimethylamine (Appendix Fig. 1B and EV4A). Application of 10-μM SEA0400 significantly blocked $Ca^{2+}$ influx by reverse activity of $NCX1.3^{WT}$ (Fig. EV4A). Furthermore, the G832 and G836 are located on the same face of the N-terminus of TM8 (Fig. 4D), close to the methxyphenol group of the SEA0400 molecule. Replacing these glycine residues with asparagine ($NCX1.3^{G832N}$ and $NCX1.3^{G836N}$) elicit comparable $Ca^{2+}$ signal to the $NCX1.3^{WT}$ (Fig. EV4B,C). In agreement with previous reports (Iwamoto et al, 2004a; Iwamoto et al, 2001), the $NCX1.3^{G832N}$ mutant is less sensitive to 10 μM SEA0400 than the $NCX1.3^{WT}$ (Figs. 4E and EV4B). In contrast, the sensitivity of $NCX1.3^{G836N}$ to 10 μM SEA0400 was almost abolished (Figs. 4E and EV4C). Based on sequence alignment, most residues involved in the SEA0400 binding are conserved among NCX1–3, except for $F248^{TM5}$, which is substituted by leucine in NCX3 (Appendix Fig S3). Interestingly, substitution of $F248^{TM5}$ by leucine indeed does not affect function of the NCX1.3, but greatly decreased the extent of inhibition by SEA0400, confirmed a previous report that $F248^{TM5}$ of NCX1 and the corresponding F242 of the NCX2 and L247 of the NCX3 play predominant roles in generating differential sensitivity of SEA0400 (Iwamoto et al, 2004b) (Figs. 4E and EV4D). The $C245^{TM5}$ is located right above the $F247^{TM5}$ and its small side chain creates enough space for SEA0400 binding (Fig. 4D). Replacement of $C245^{TM5}$ with a bulky phenylalanine ($NCX1.3^{C245F}$) has no effect on transport activity, but abolish its sensitivity to 10 μM SEA0400 (Figs. 4E and

EV4E). We also found that the $NCX1.3^{\Delta XIP}$ mutant markedly reduced sensitivities to inhibition by SEA0400, in line with the previous study suggesting that the XIP region is also important in conferring the sensitivity to SEA0400 in NCX1 (Iwamoto et al, 2004b) (Figs. 4E and EV4F). However, no directly interactions are identified between the SEA0400 molecule and the XIP in our structure. Structural comparison of previous determined inward-facing CAXs homologues (Nishizawa et al, 2013; Waight et al, 2013; Wu et al, 2013b) revealed that the TM2ab exhibits substantial conformational heterogeneity (Fig. EV5B). Among these conformations, TM2 showed less bending in NCX1.3, likely due to the extensive interactions between Loop1-2 and the XIP element that indirectly stabilize TM2ab in its current position. Importantly, this stabilization is critical for SEA0400 binding, as SEA0400 molecule forms extensive interactions with TM2ab (Fig. EV1D). We speculate that the enhanced sensitivity of SEA0400 to NCX1.3 via XIP is due to the fact that XIP stabilizes TM2ab of $NCX1.3^{IF}$ at a certain conformation that preferentially binds with SEA0400, and thus the XIP and SEA0400 synergistically inhibit $Ca^{2+}$-uptake activity of NCX1.3. In current NCX1.3, the interactions between the XIP and β sheet preceding the TM2ab would stabilize its conformation (Fig. EV3A,B). We speculate that the enhanced sensitivity of SEA0400 to NCX1 via XIP is due to the fact that XIP stabilizes TM2ab of $NCX1.3^{IF}$ at a certain conformation that preferentially binds with SEA0400, and thus the XIP and SEA0400 synergistically inhibit $Ca^{2+}$-uptake activity of NCX1.

In the outward-facing structure of mjNCX, the N-terminal part of TM2 (TM2ab) breaks into two short helices (TM2a and TM2b) due to a π-bulge occurring at A43 (Liao et al, 2012). However, in the presence of SEA0400, TM2ab of NCX1.3 resumes a canonical α-helix. Considering the close contacts between SEA0400 and TM2ab, we speculate that SEA0400 may not prefer to bind with NCX1.3 in the presence of a bent TM2ab. The bent conformation of TM7ab is believed to be critical for the conformational change from outward-facing to inward-facing state, and all of the outward-facing structures are featured by a bent TM2ab (Liao et al, 2016). We thus postulate that a similar bent in TM2ab is important for the conformational transition from the inward-facing to outward-facing state. In agreement, since SEA0400 stabilizes the straight α-helical conformation and disrupts the local backbone dynamics of TM2ab, the association of SEA0400 allosterically inhibits the conformational transition. Moreover, the bending angle between TM2ab and TM2c is ~135° in NCX1.3, which is smaller than that in the $mjNCX^{OF}$ structure as well as $NCX1.3^{OF}$ model (Figs. 4F and EV2B). Consequently, in the $mjNCX^{OF}$ structure, TM2ab bends towards the C-terminal part of TM8 and forms contacts with it. However, such contacts are prohibited by the bound SEA0400 in

NCX1.3, since the space between TM2ab and TM8 is now occupied by SEA0400 (Fig. 4F). It is further anticipated that the smaller bending angle in TM2 caused by SEA0400 binding will also hamper conformational transition from the inward-facing to the outward-facing state, and thus in part contribute to the inhibitory effect of SEA0400.

## Discussion

The sodium-calcium exchanger (NCX) can mediate the exchange of $Na^+$ and $Ca^{2+}$ in both forward and reverse transport modes, regulating the $Ca^{2+}$ homeostasis in cells and participating in various physiological and pathological processes. In this study, we elucidated the cryo-electron microscopy (cryo-EM) structure of the inward-facing conformation of human NCX1.3 in the presence of the SEA0400-specific inhibitor. The XIP region, responsible for mediating $Na^+$-dependent inactivation, is positioned within a groove situated between TMD and CBD2, stabilizing the inward-facing conformation and preventing a transition to the outward-facing state by clashing with gating helices TMs$^{1/6}$ at the outward-facing state (Fig. 3C). However, due to limited resolution, the precise localization of $Na^+$ within the structure could not be determined, preventing the elucidation of its molecular mechanism in facilitating XIP binding. Cytoplasmic $Ca^{2+}$ acts not only as a substrate for the $Na^+/Ca^{2+}$ exchanger, but also regulates its activity of NCX by binding to the intracellular domains, CBD1 and CBD2. CBD1 contains four $Ca^{2+}$ binding sites ($Kd_{Ca1/Ca2}$: ~10 μM, $Kd_{Ca3/Ca4}$: ~0.4 μM) (Hilge et al, 2009a). The number of $Ca^{2+}$ binding sites in CBD2 is determined by its mutually exclusive exons, exon A and exon B. The CBD2 with exon A has two $Ca^{2+}$ binding sites ($Kd_{Ca1}$:9 μM, $Kd_{Ca2}$:1 μM) (Hilge et al, 2009a), whereas CBD2 with exon B lacks $Ca^{2+}$ binding sites (Hilge et al, 2009a). The NCX1.3 contains exon B and thus we speculate that only CBD1 may bind with $Ca^{2+}$ in current structure. Previous investigations showed that the binding of $Ca^{2+}$ to CBD1 activates the transport activity of the NCX protein (Khananshvili, 2020) and the presence of $Ca^{2+}$ leads to the stabilization of a specific conformation of CBD1 and CBD2 (Giladi et al, 2012). However, when $Ca^{2+}$ dissociates, CBD1 loses its structural integrity (Hilge et al, 2006). The association of $Ca^{2+}$ with CBD1 has been also reported to have varying effects on the conformation of CBD2 in different splicing isoforms, thereby exerting distinct modulatory effects on the transport activity (Khananshvili, 2020). Nevertheless, we have been unable to reach a conclusive understanding of how $Ca^{2+}$ bound to CBD1 activates the transporter allosterically, primarily due to the limitation of having only one structure that is potentially bound with $Ca^{2+}$. Further investigations and additional structures are needed to shed more light on this matter. Despite CBD2 of NCX1.3 being unable to bind with $Ca^{2+}$, structural analysis reveals that the potential $Ca^{2+}$ binding site of CBD2 with exon A is located in close proximity to the XIP binding site (Fig. 3C). In the structure of NCX1.3, CBD2 exhibits a negatively charged electrostatic surface, which facilitates the binding of the positively charged XIP region with the TMD (Fig. 3C). We further hypothesize that the binding of $Ca^{2+}$ to CBD2 with exon A would neutralize this negative electrostatic potential, consequently reducing the binding affinity of the XIP region. This, in turn, could result in a reduction or even complete elimination of XIP-mediated inactivation.The selective inhibitor SEA0400 predominantly blocks NCX1 and exerts almost no influence on NCX3

(Iwamoto et al, 2004b). In our structure, it binds to an allosteric binding site where most of the residues are conserved, except for F248, which is substituted with L in NCX3 (Fig. 4C). Functional experiments further confirm the crucial role of this residue in subtype selectivity (Fig. 4E), which provides a solid foundation for the rational design of subtype-specific drugs. Previous investigations have demonstrated that the SEA0400 is preferable for inhibiting the reverse mode of NCX (Tanaka et al, 2002), which is responsible for the uptake of extracellular $Ca^{2+}$. However, the molecular basis for its selectivity towards the reverse mode over the forward mode remains unknown. Understanding this molecular basis is crucial for rational development of pharmacological inhibitors that specifically target NCX1 in the reverse mode, holding promising therapeutic potential in preventing $Ca^{2+}$ overload and electrical dysfunction in the context of ischemia/reperfusion injury and heart failure (Chen and Li, 2012a; Imahashi et al, 2005). When the NCX operates in the reverse mode, the high concentration of intracellular $Na^+$ plays a dual role. On one hand, it is required to act as a driven ion, while on the other hand, it facilitates the inactivation of NCX1.3 through the XIP region, thereby effectively preventing cells from $Ca^{2+}$ overloading by NCX during the pathological $Na^+$ accumulation (He et al, 2000; Hilgemann et al, 1992; Li et al, 1991; Matsuoka et al, 1993). As the sample is purified in the presence of high concentration $Na^+$, we speculate that the complex structure may represents an inactive state, primarily associated with the reverse mode. The XIP engages with the TM1-2 loop and plays a crucial role in the inactivation process by stabilizing the inward-facing conformation and preventing a transition to the outward-facing state. Furthermore, these interactions between the XIP and TM1-2 loop also maintain a specific conformation of TM2ab, which is essential for the allosteric binding of SEA0400. Our calcium imaging experiments have also demonstrated that the XIP is vital for the sensitivity of NCX to SEA0400 (Figs. 4E and EV4F). In the forward mode, we speculate that the stability of the SEA0400 binding pocket is compromised compared to the reverse mode, because the XIP may not effectively interact with the TM1-2 loop at lower intracellular $Na^+$, supported by the minimal inactivation observed in the forward mode over time (Iwamoto et al, 2004a; Matsuoka et al, 1995), thereby failing to stabilize the SEA0400 binding pocket. Hence, we hypothesis that the $Na^+$-dependent XIP-mediated inactivation, which is more prominently observed in the reverse mode, leads to a more stable SEA0400 binding pocket compared to the forward mode, resulting in the selective inhibition of the reverse mode by the SEA0400. In summary, our research demonstrates that in the reverse mode, the high concentration of intracellular $Na^+$ promotes XIP interactions with TMD and CBD2, leading to inactivation of NCX1.3. This process also stabilizes the allosteric binding pocket of SEA0400, underlying the molecular basis for the synergistic inhibition of NCX1.3 by XIP and SEA0400, with SEA0400 selectively targeting NCX1.3 in the reverse mode.

## Methods

### Fura-2 $Ca^{2+}$ imaging

HEK293T cells (Gibco, USA) were seeded on 25-mm round glass coverslips (NEST) coated with poly-L-lysine (Sigma-Aldrich),

which were grown in 35-mm dishes and cultured in DMEM supplemented with 10% fetal bovine serum (FBS) at 37 °C and 5% $CO_2$ in an incubator. HEK293T cells were transiently transfected with 3 µg of plasmid DNA encoding NCX1.3 using Lipofectamine 2000™ (Invitrogen) according to the manufacturer's recommended procedure. For microfluorimetric studies, HEK293T cells were used at 24 h after transfection and washed thrice with Krebs' buffer (5.5 mM KCl, 160 mM NaCl, 1.2 mM $MgCl_2$, 1.5 mM $CaCl_2$, 10 mM glucose, and 10 mM HEPES-NaOH pH 7.4). Subsequently, HEK293T cells were incubated with 2.5 µM $Ca^{2+}$ Indicator dyes Fura 2-AM (Sigma-Aldrich) and 0.05% Pluronic F-127 (Beyotime Biotechnology) at 37 °C for 30 min in Krebs' buffer. Moreover, $Na^+/K^+$-ATPase inhibitor ouabain (2 µM) was included throughout the $Ca^{2+}$ measurement, and then Fura 2-AM loaded cells were washed triple times with Krebs' buffer. The coverslips with cells were mounted on a perfusion chamber and placed onto the stage of an inverted fluorescence microscope (IX81, Olympus) equipped with a Fluar 40× oil objective lens and a cooled CCD camera (Zyla sCMOS, Andor). To evaluate NCX1.3 activity of $[Na^+]_{in}$-dependent $Ca^{2+}$ uptake through the reverse mode, Krebs' buffer was changed to $Na^+$-free $NMDG^+$ buffer (5.5 mM KCl, 147 mM N-methylglucamine, 1.2 mM $MgCl_2$, 1.5 mM $CaCl_2$, 10 mM glucose, and 10 mM HEPES-NaOH, pH 7.4) by the perfusion system during calcium imaging. Finally, 500 µM ATP was added to as a control to confirm the cells we monitored are living cells (Xiao et al, 2011). The Fura 2-AM fluorescence intensity was excited at 340 nm and 380 nm by a high-speed wavelength switcher (Lambda DG-4; Sutter Instrument Co.), and 500 nm emission light images were captured every 300 ms. At least 20–30 individual cells with mCherry fluorescence were monitored and analyzed in each experiment. Intracellular $Ca^{2+}$ concentration of NCX1.3 was indicated as the 340/380 ratio by using Andor software (Andor iQ3.6.3, Andor). To determine the inhibitory effect of SEA0400 on NCX1.3 activity, 10 µM SEA0400 was incubated for 30 min when Fura 2-AM was loaded and was maintained during the $Ca^{2+}$ imaging. Data were acquired with Andor software and analyzed with GraphPad Prism 9 software (GraphPad software) and were expressed as means ± SEM.

## Whole-cell voltage-clamp recordings

$I_{NCX}$ recording referred to published method previously (Dong et al, 2002). $I_{NCX}$ outward currents were carried out by electrophysiological measurements in whole-cell configuration with HEK 293T cells at room temperature. Coverslips with attached transfected cells were placed in the chamber within bath solution containing 145 mM LiCl, 1 mM $MgCl_2$, 10 mM D-glucose, 10 mM HEPES, pH 7.4, or 1 mM $CaCl_2$. Pipettes were prepared from a Sutter P-97 puller and fire-polished to a resistance of 2–4 megohms (Narishige MF-830 microforge) when filled with solution. The pipette solution contained 120 mM NaCl, 5 mM KCl, 2 mM $MgCl_2$, 20 mM tetraethylammonium-chloride (TEA-Cl), 1 mM $Na_2ATP$, 10 mM HEPES, 8 mM D-glucose, pH 7.2, 5 mM EGTA plus 4.28 mM $CaCl_2$, which generated a free $[Ca^{2+}]$ of 1 µM. The current recording was conducted with Axoclamp 700B amplifier and Digidata 1440 A (Axon Instruments, Inc), filtered at 10 kHz. Cells were voltage-clamped at a holding potential of 0 mV and induced $I_{NCX}$ outward currents from $Ca^{2+}$-free to 1 mM $CaCl_2$ within bath solution. Data analyses were performed using Clampfit, Excel 2016 and GraphPad Prism 9 software (GraphPad software).

## CPM-based thermostability assay

To evaluate the stability of the purified NCX1.3 transporter in the presence or absence of SEA0400, thermal stability assays were used as described previously (Alexandrov et al, 2008). Thermofluor utilizes the binding of a dye called N-[4-(7-diethylamino-4-methyl-3-coumarinyl) phenyl] maleimide (CPM; Invitrogen), which reacts with thiol groups of cysteine residues by maleimide reaction chemistry. The purified NCX1.3 protein sample with a concentration of 6 mg/mL was finally diluted to a final concentration of 0.2 mg/mL in a total volume of the 25 µL assay buffer (20 mM HEPES, pH 7.5, 150 mM NaCl, 0.007% (w/v) GDN). Freshly made CPM was added to the assay buffer by diluting the stock (4 mg/mL in DMSO) 16-fold, resulting in a solution of 1 µL of CPM per 25 µL PCR tube. For the thermal stability assays investigating the NCX1.3 protein in the presence of SEA0400, the reaction mixture was supplemented with 100 µM SEA0400 and further incubated for 15 min on ice in the dark. Thermal denaturation was performed using a RotorGene 6600 (Qiagen, USA) real-time PCR instrument with a ramp rate of 1.0 °C/min and a step-wise temperature increase from 25 to 95 °C in 1.0 °C/cycle increments (ramp rate of 1.0 °C/min) and with an equilibration time of 5 s at each temperature. The fluorescence signal was measured using filters with an excitation wavelength of 384 nm and an emission wavelength of 470 nm. Data were exported for analysis in Microsoft Excel 2016 and analyzed with GraphPad Prism 9 software (GraphPad software).

## Expression and purification of human NCX1.3

The gene of the full-length human NCX1.3 (UniProtKB accession: P32418-2) was amplified from a human cDNA library using the Super-Fidelity DNA Polymerase (Vazyme). The NCX1.3 gene was subcloned into a modified pEG BacMam vector. A Twin-Strep tag and mCherry followed by a PreScission Protease (PPase) cleavage site were inserted after the N-terminus of the NCX1.3 signal peptide (amino-acid residues 1−35). The NCX1.3 protein was expressed in HEK293F cells using the Bac-to-Bac baculovirus expression system (Invitrogen, USA). P1 and P2 viruses of NCX1.3 were produced in sf9 insect cells (Gibco, USA) for subsequent protein overexpression. The HEK293F cells (Gibco, USA) were infected by the P2 viruses at a density of $2.5 \times 10^6$ cells/mL. The cells were cultured at 37 °C in suspension supplemented with 1 µM inhibitor SEA0400 (MedChemExpress, USA), 1% (v/v) fetal bovine serum, and 5% $CO_2$ in a shaking incubator. After 12 h, 10 mM sodium butyrate was added into the culture. Subsequently, the cells were harvested after another 48 h and were stored at −80 °C immediately after being frozen in liquid nitrogen.

The HEK293F cells expressing NCX1.3 were resuspended in a purification buffer A [20 mM HEPES pH 7.5, 150 mM NaCl, 1 mM $CaCl_2$, 5 mM β-mercaptoethanol (β-ME), 1 µM SEA0400, aprotinin (2 µg/mL), leupeptin (1.4 µg/mL), and pepstatin A (0.5 µg/mL)]. The cellular membrane was then broken using a Dounce homogenizer and collected by centrifugation at $100,000 \times g$ for 1 h and solubilized by addition of 1% (w/v) Lauryl Maltose Neopentyl Glycol (LMNG, Anatrace), 0.2% (w/v) cholesteryl hemisuccinate (CHS, Anatrace), 2 mM ATP, and 5 mM $MgCl_2$ at 4 °C for 2 h with rotation. Subsequently, the insoluble cell debris was removed by centrifugation at $100,000 \times g$ for 1 h. The

supernatant was filtered and passed through Streptactin Beads (Smart-Lifesciences, China) pre-equilibrated with the wash buffer B [20 mM HEPES pH 7.5, 150 mM NaCl, 5 mM β-ME, 1 mM CaCl$_2$, 1 μM SEA0400, and 0.01%(w/v) glycodiosgenin (GDN, Anatrace)]. The beads were washed with 10 column volumes of wash buffer B supplemented with 2 mM ATP and 5 mM MgCl$_2$ to remove nonspecific bound protein. The NCX1.3 sample was eluted with the wash buffer B supplemented with 5 mM desthiobiotin. The eluted protein was subsequently digested with PPase at 4 °C for 3 h with gentle rotation. The digested protein sample was concentrated in the 100 kDa MW cut-off spin concentrators (Merck Millipore, Germany) and further purified by gel filtration (Superose 6 Increase 10/300 GL, GE Healthcare, USA) pre-equilibrated in the buffer C [20 mM HEPES pH 7.5, 150 mM NaCl, 5 mM β-ME, 1 mM CaCl$_2$, and 0.01%(w/v) glycodiosgenin (GDN, Anatrace)]. The peak fractions were pooled and concentrated to 6 mg/mL for cryo-EM sample preparation. For NCX1.3 sample preparation of the drug-bound complex, 400 μM SEA0400 was supplemented and incubated with protein sample for 30 min on ice before freeze grids.

## Cryo-EM sample preparation and data acquisition

Holely carbon grids (Cu R1.2/1.3 300 mesh, Quantifoil) were glow-discharged in H$_2$-O$_2$ condition for 1 min using the Solarus plasma cleaner (Gatan, USA). A droplet of 2.5 μL of purified NCX1.3 protein sample (6 mg/mL) was applied on glow-discharged holey carbon grids. The grids were then automatically blotted for 4 s at 4 °C under 100% humidity using a Vitrobot Mark IV (Thermo Fisher Scientific, USA) and vitrified in liquid ethane. The grids were then transferred to a Titan Krios G2 operating at 300 kV, and cryo-EM data were collected by a K2 summit direct electron detector with a GIF quantum LS energy filter. The slit width of the energy filter was set to 20 eV. Movie stacks were collected using SerialEM at the magnification of ×165,000. The pixel size on motion-corrected micrographs was 0.82 Å. The dose rate and exposure time were set to ~10.0 Å/pixel*s and ~4 s, respectively. Each movie stack was dose-fractioned in 32 frames with a total dose of ~ 60 e$^-$/Å$^2$ on the image. The defocus range was set to the range between −1.2 μm and −2.2 μm.

## Cryo-EM data analysis

For the dataset of NCX1.3 bound with SEA0400, 12,573 movie stacks were collected. The beam-induced motion was corrected by MotionCor2 (Zheng et al, 2017)), and contrast-transfer function (CTF) parameters were estimated by Gctf (Zhang, 2016). Micrographs were filtered based on the detected CTF estimation resolution better than 6 Å followed, and 12,134 micrographs were subjected to particle picking in cryoSPARC (Punjani et al, 2017). A total of 2,953,234 particles were picked and used to three rounds of 2D classification. For each round of 2D classification, only classes featuring transmembrane helices and discernible soluble domain were selected. Subsequently, 469,943 particles were used for ab initio reconstruction (initial alignment resolution of 12 Å, maximum resolution of 6 Å, no class similarity) and generated six classes. One of six initial models exhibited both typical transmembrane helix densities and distinguishable soluble domain densities, which was set as a good reference map for the subsequent heterogeneous refinements. Rounds of heterogeneous refinements were employed to sort 469,943 particles yielding 194,744 good

particles. After non-uniform refinement, this dataset gave rise to a 4.7-Å map clearly showing densities of transmembrane helices. In order to improve map resolution, seed-facilitated 3D classification was performed to obtain more good particles from particle sets picked by Blob picker and Template picker. After that, 365,313 particles were selected and subjected to non-uniform refinement. The final map was reported at 3.5-Å resolution according to the Gold-standard Fourier shell correlation criterion (Scheres, 2012).

## Model building

The 3.5-Å high-resolution cryo-EM map of NCX1.3 clearly showed densities of main chains and side chains of transmembrane region and a part of the soluble region near the membrane, which allow us to reliably build and adjust the model of these parts. The core domain (TM2-TM5 and TM7-TM10) of crystal structure of mjNCX (PBD ID: 3V5S) (Liao et al, 2012) was extracted using PyMOL software and then docked into the map as a rigid body using UCSF Chimera (Pettersen et al, 2004) to roughly determine the arrangement and connection of these transmembrane helices. Each helix was further refined and adjusted based on the map using COOT. In addition, the TM1 and TM6 helices were built manually in COOT based on the density map. Finally, we conducted careful manual adjustments based on the density of amino acid side chains, guided by well-resolved densities of residues with bulky side chains. The intracellular regions closed to the plasma membrane were manually built guided by the well-resolved densities of residues with bulky side chains such as tryptophan, phenylalanine, and arginine. To build an intact model of NCX1.3, CBD1 (PDB ID: 2DPK) (Nicoll et al, 2006b) and CBD2 (PDB ID: 2KLT) (Hilge et al, 2009b) structures were subsequently rigid-body fitted into the 3.5-Å cryo-EM with well-defined densities of most main-chains of CBD1 and CBD2 in UCSF Chimera, followed by manually refinement against the corresponding cryo-EM maps in Coot. The structure data files (SDFs) of SEA0400 were obtained from PubChem, followed by the generation of 3D models and refinement restraints in phenix.ligand_eLBOW (Adams et al, 2010). The SEA0400 molecule was docked in the EM map and refined according to the corresponding density. Then, the overall model was refined against the cryo-EM map by using the phenix.real_space_refine program in the presence of secondary structure restraints, and the resulting model stereochemistry was evaluated using the comprehensive validation (cryo-EM) utility in PHENIX (Adams et al, 2010).

All structural Figs were generated using UCSF Chimera, ChimeraX (Goddard et al, 2018), and Pymol (DeLano, 2002).

## Molecular dynamics simulations

We created an alternate model by reversing the orientation of SEA0400 (NCX1.3$^{rev}$), specifically flipping it upside down compared to its original positioning in the cryo-EM structure. In this model, the difluorophenyl group was inserted deeply into the binding pocket, while the 3-ethosyanilion group was directed toward the intracellular side. The cryo-EM structure of SEA0400-bound NCX1.3 (NCX1.3$^{EM}$) and model of NCX1.3$^{rev}$ were used as the initial structure for molecular dynamics simulation. In modeling process, due to the intracellular structural domains CBD1 and CBD2 were not implicated in SEA0400 binding, we removed them, and retained the residues 51–283, 683–720, and 736–937 for subsequent analysis. The molecular dynamics simulations were

conducted using GROMACS v.2021.6 (Abraham et al, 2015) with the Amber ff14SB force field (Maier et al, 2015) for protein and lipids and GAFF2 (Wang et al, 2006) for SEA0400. All the input files were generated by CHARMM-GUI (Jo et al, 2008). The cryo-EM structure NCX1.3$^{EM}$ and model of NCX1.3$^{rev}$ were separately embedded into a pre-equilibrated 1-palmitoyl-2-oleoyl-sn-glycero-3-phosphatidylcholine membrane (POPC) with the membrane orientation calculated by the Orientations of Proteins in Membranes database (PPM 2.0) (Lomize et al, 2012). All systems were minimized for 5000 steps. Equilibrations with weak restraints were conducted before running production MD, following standard CHARMM-GUI membrane equilibration steps. The configuration of the simulation box was chosen as rectangular. The system was then solvated with TIP3P water (Jorgensen et al, 1983), and 150 mM NaCl was incorporated to neutralize the system using the Monte Carlo method. In the final NCX1.3$^{EM}$ system, there were a total of 62,973 atoms, including 113 POPCs, 13418 water molecules, 73 counter ions, the transmembrane region of NCX1.3$^{EM}$, and SEA0400$^{EM}$. The final NCX1.3$^{rev}$ system contained 62,756 atoms in total, which consisted of 114 POPCs, 13,301 water molecules, 73 counter ions, the transmembrane region of NCX1.3$^{rev}$, and SEA0400$^{rev}$. In each system, three separate 100 ns unrestrained production simulations were conducted, maintaining a constant particle number, 1 bar pressure, and 303.15 K temperature. The velocity-rescaling thermostat (Bussi et al, 2007) and Parrinello-Rahman barostat (Parrinello and Rahman, 1980) were employed for temperature and pressure coupling, respectively. Long-range electrostatic interactions were calculated using the particle mesh Ewald (PME) method. A cut-off of 12 Å was established for van der Waals interactions, incorporating a force-switching function at 10 Å. The LINES algorithm (Hess, 2008) was utilized to keep bonds involving hydrogen fixed during each 2 fs integration time step. Clustering analysis was carried out using the GROMOS software (Daura et al, 1999).

## Data availability

The three-dimensional cryo-EM density map of the SEA0400 bound NCX1.3 complex has been deposited in the Electron Microscopy Data Bank under the accession code EMD-36465. The coordinate for the SEA0400 bound NCX1.3 complex has been deposited in Protein Data Bank under accession code 8JP0.

## Peer review information

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

## Acknowledgements

We thank X. Huang, B. Zhu, X. Li, and other staff members at the Center for Biological Imaging (CBI), Core Facilities for Protein Science at the Institute of Biophysics, Chinese Academy of Science (IBP, CAS) for the support in cryo-EM data collection. We thank Y. Teng and Y. Feng from the Center for Biological Imaging (CBI), Institute of Biophysics, Chinese Academy of Science for their help of the calcium imaging experiment. We thank B. Xiao, Y. Jiang, and W. Liu from Tsinghua University for their help of the calcium imaging experiment. We thank J. Hu and K. Li from the Institute of Biophysics, Chinese Academy of Science, for the helpful discussion about the calcium imaging experiment. We thank Dr. Xuejun Cai Zhang for critical reading of the manuscript. Yan Wu for his research assistant service. This work was supported by the Chinese National Programs for Brain Science and Brain-like Intelligence Technology (grant 2022ZD0205800 to YZ), the Chinese Academy of Sciences Strategic Priority Research Program (grant XDB37030304 to YZ), the National Natural Science Foundation of China (grant 92157102 to YZ, grant 32301026 to YD, and grant 32200978 to LH), the National Key Research and Development Program of China (grant 2021YFA1301501 to YZ), China Postdoctoral Science Foundation, and the Youth Innovation Promotion Association of the Chinese Academy of Sciences (grant 2022089 to LH).

## Author contributions

**Yanli Dong**: Data curation; Formal analysis; Investigation; Methodology; Writing—original draft. **Zhuoya Yu**: Data curation; Software; Formal analysis; Investigation; Writing—original draft. **Yue Li**: Data curation; Software; Investigation; Methodology; Writing—original draft. **Bo Huang**: Software; Formal analysis; Methodology; Writing—original draft. **Qinru Bai**: Software; Formal analysis; Methodology; Writing—original draft. **Yiwei Gao**: Software; Formal analysis; Methodology. **Qihao Chen**: Software; Methodology. **Na Li**: Investigation; Methodology. **Lingli He**: Investigation; Methodology. **Yan Zhao**: Conceptualization; Formal analysis; Supervision; Funding acquisition; Validation; Investigation; Visualization; Methodology; Writing—original draft; Project administration.

## Disclosure and competing interest statement

The authors declare no competing interests.

# Expanded View Figures

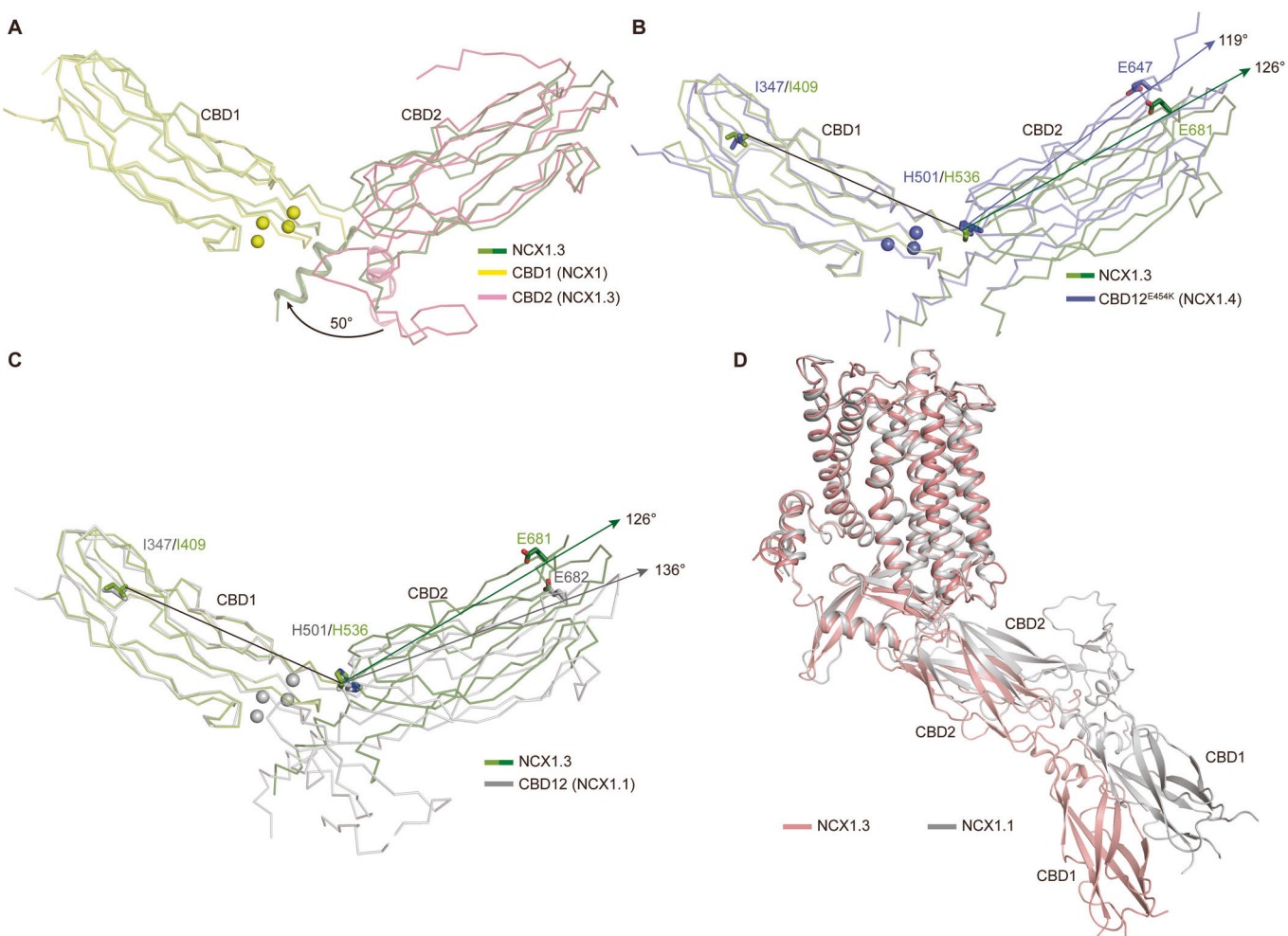

**Figure EV1. Structural alignment of CBDs among human NCX1 isoforms.**

Related to Fig. 1. (A) Structural comparison of CBDs between NCX1.3 and NCX1. The $Ca^{2+}$-bound CBD1 structure of NCX1 (PDB ID: 2DPK) is colored in yellow and the bound $Ca^{2+}$ ions are displayed as yellow spheres. CBD1 and CBD2 of NCX1.3 in our structure are colored in green and dark green, respectively. The CBD2 of NCX1.3 (PDB ID: 2KLT) is colored in pink. (B) Structural comparison of CBDs between NCX1.3 and NCX1.4 (CBD12$^{E454K}$, PDB ID: 3US9) using the invariant CBD1 domains. CBD1 and CBD2 of NCX1.4 are colored in purple. The key residues are shown as sticks. The bound $Ca^{2+}$ ions of CBD1 for NCX1.4$^{E454K}$ are displayed as purple spheres. The angles between CBD1 and CBD2 of NCX1.3 and NCX1.4 are marked, respectively. (C) Structural comparison of CBDs between NCX1.3 and NCX1.1 (PDB ID: 8SGJ). CBD1 and CBD2 of NCX1.3 in our structure are colored in green and dark green, respectively. (D) Structure superposition of NCX1.3 and NCX1.1 (PDB ID: 8SGJ) using TMD domain. NCX1.3 and NCX1.1 are shown as salmon and gray cartoon, respectively.

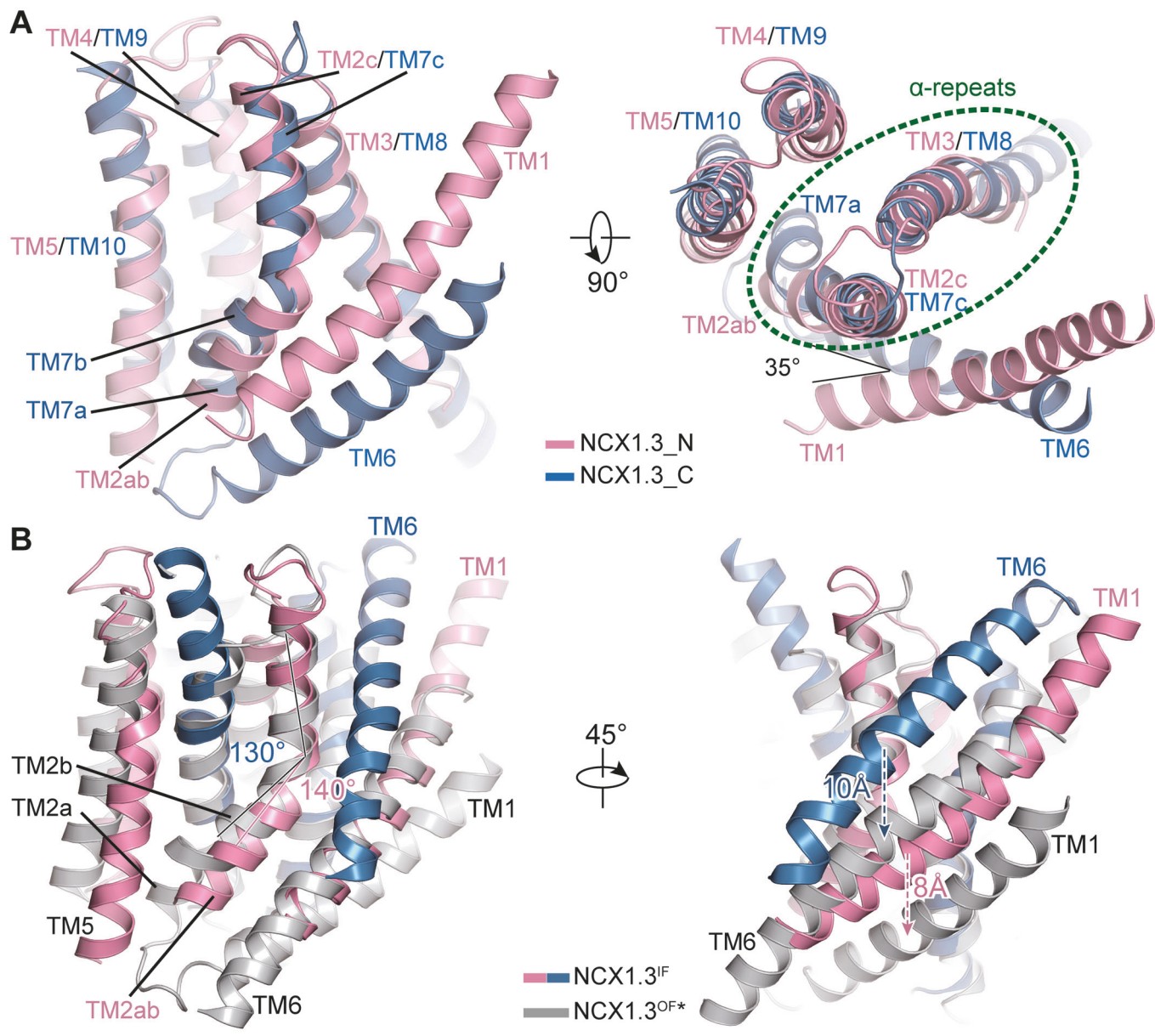

**Figure EV2. Structural comparison of NCX1.3 between N-terminal and C-terminal.**

Related to Fig. 2 and Fig. 4. (A) Structural comparison between NCX1.3_N (N-terminal half of NCX1.3, in pink) and inverted NCX1.3_C (C-terminal half of NCX1.3, in dark blue) viewed from parallel to the membrane and the extracellular side, respectively. The α-repeats are marked as the green dotted oval. The angle between TM1 and TM6 of NCX1.3_N is indicated. (B) Structural comparison between NCX1.3 (in pink and dark blue) and outward facing NCX1.3 model (NCX1.3$^{OF*}$, in gray) model using the core domain (TM2–5 and TM7–10) viewed parallel to the membrane and with rotation of 45°. NCX1.3$^{OF*}$ model is the inverted structure of NCX1.3$^{IF}$ due to the symmetry of NCX1.3 structure and its ion-binding sites. TM1 to TM5 were superposed on TM6 to TM10 in an analogous manner for mjNCX. The helix bending between TM2ab and TM2c of NCX1.3 and NCX1.3$^{OF*}$ are indicated, respectively. The displacements of TM1 and TM6 between NCX1.3 and NCX1.3$^{OF*}$ are indicated.

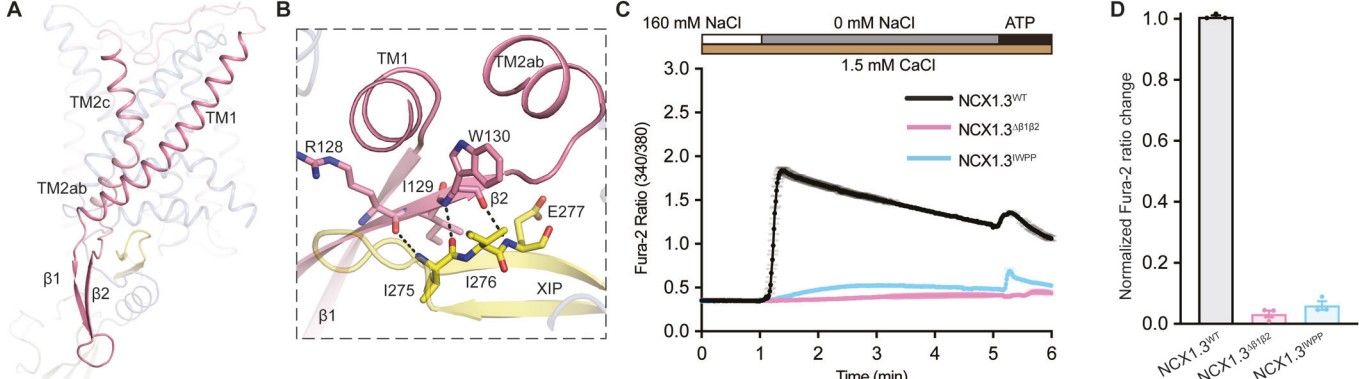

**Figure EV3. Structural and functional characterization of β sheets between TM1 and TM2 in NCX1.3.**

Related to Fig. 3. (A) The position of β1 and β2 in the overall structure. TM1, β1, β2 and TM2 are shown as pink cartoons. (B) Detailed binding sites between β2 and XIP region. The side chains of key residues are displayed in sticks and black dashed lines indicate potential hydrogen bonds. (C) Representative averaged traces of cytosolic $Ca^{2+}$ measurements for NCX1.3$^{WT}$, NCX1.3$^{\Delta\beta1\beta2}$ and NCX1.3$^{IWPP}$. (D) Normalized Fura-2 ratio change values are measured as the difference between the initial ratio values and the peak ratio values for NCX1.3$^{WT}$ ($n = 3$), NCX1.3$^{\Delta\beta1\beta2}$ ($n = 3$) and NCX1.3$^{IWPP}$ ($n = 3$).

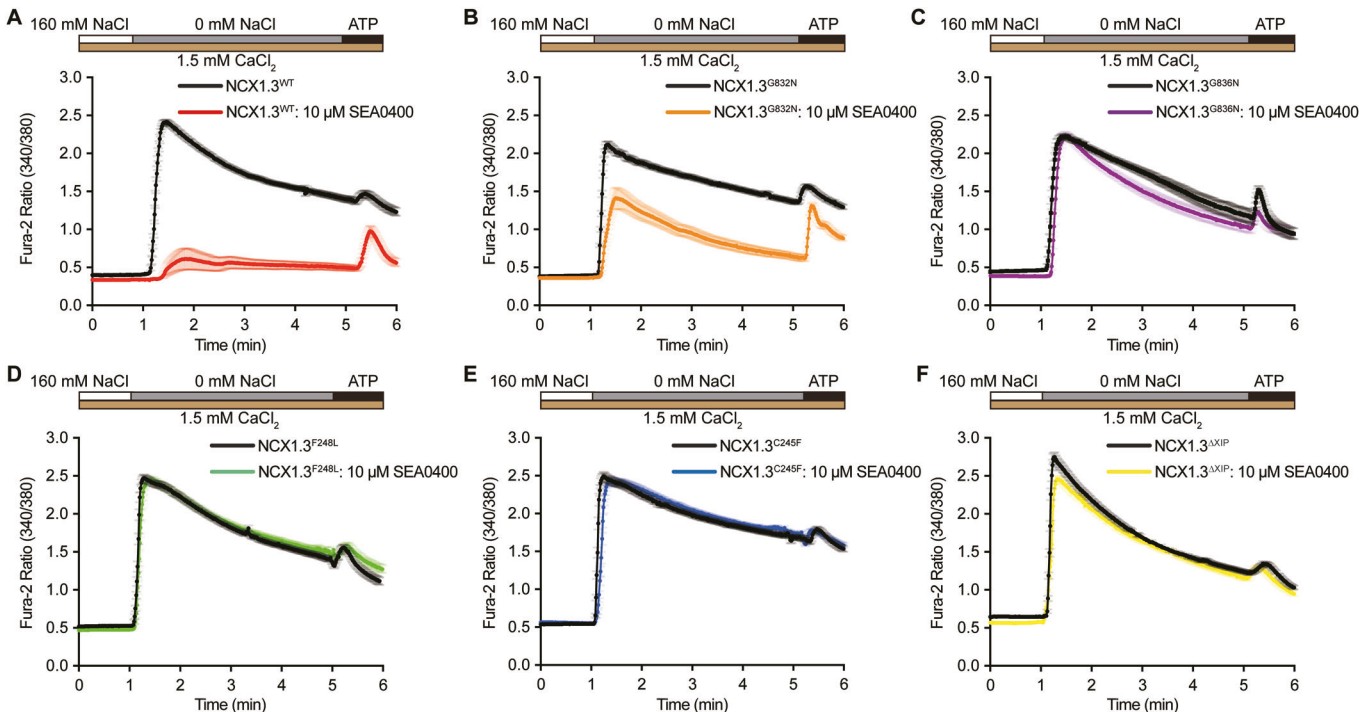

**Figure EV4.  Effect of SEA0400 on NCX1.3 activity in HEK293 cells.**

Related to Fig. 1, Fig. 3, and Fig. 4. (**A–F**) Representative averaged traces of cytosolic $Ca^{2+}$ ($[Ca^{2+}]_i$) measurements showing $[Ca^{2+}]_i$ increase in HEK293 cells of wild-type and mutant variants by calcium imaging experiment. The HEK293T cells are transfected with NCX1.3$^{WT}$ (**A**), NCX1.3$^{G832N}$ (**B**), NCX1.3$^{G836N}$ (**C**), NCX1.3$^{F248L}$ (**D**), NCX1.3$^{C245F}$ (**E**) and NCX1.3$^{\Delta XIP}$ (**F**). These mutants are designed to disturb SEA0400 binding pocket of NCX1.3. Each panel depicts superimposed traces representing the $[Ca^{2+}]_i$ response when $Na^+$-free NMDG$^+$ buffer alone, or in the presence of SEA0400 (10 μM). The change of solution was marked by time breaks. The Krebs' buffer including 160 mM NaCl was changed $Na^+$-free NMDG$^+$ buffer (0 mM NaCl) at one minute. Subsequently, 500 μM ATP was added at the end as a control to measure cell viability. The Fura2 ratio (340/380) was used as a quantitative indicator of intracellular $[Ca^{2+}]$. For each experiment, data shown are representative of at least three experiments with >20 cells for each condition. Error bars are defined as mean ± SEM.

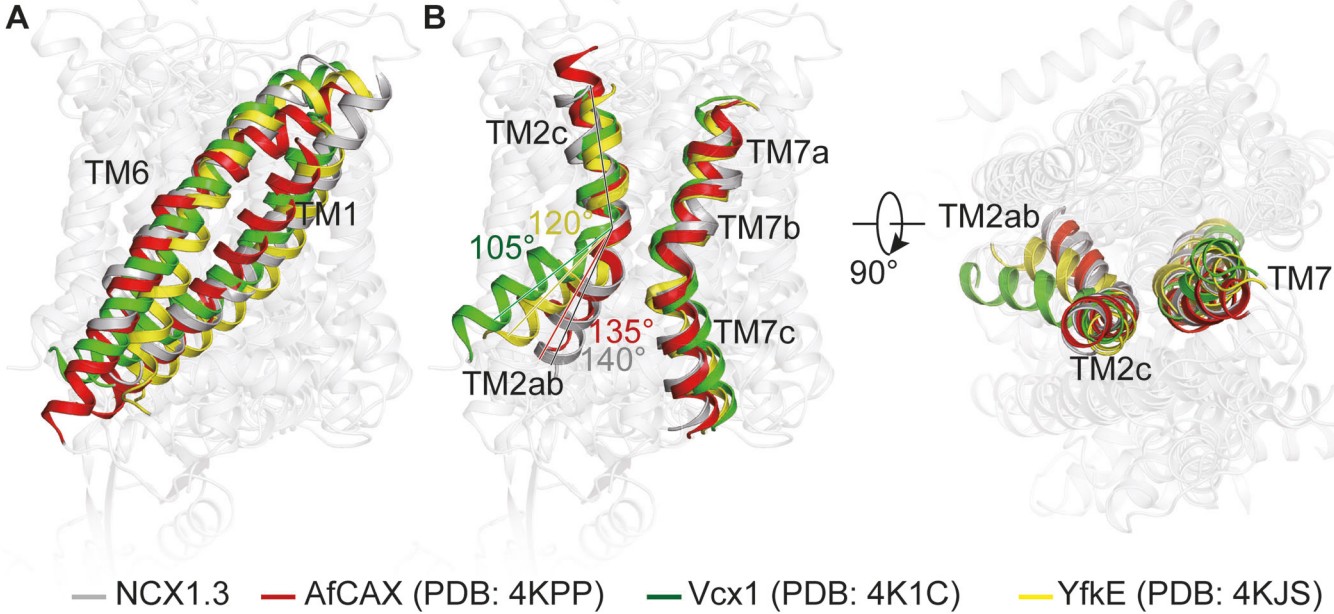

— NCX1.3    — AfCAX (PDB: 4KPP)    — Vcx1 (PDB: 4K1C)    — YfkE (PDB: 4KJS)

**Figure EV5.   Superimposition of the core domain (TM2-5 and TM7-10) among NCX1.3, AfCAX, Vcx1, and YfkE.**

Related to Fig. 4. (A) Structural comparison among NCX1.3 (in gray), AfCAX (in red), Vcx1 (in green) and YfkE (in yellow) using the core domain (TM2−5 and TM7−10). Structural changes of the TM1 and TM6 helices are shown. (B) Structural changes of the TM2 and TM7 helices. The angles of helix bending between TM2ab and TM2c among them are indicated.

