## [Peer Review File · The EMBO Journal]

Structural insight into the allosteric inhibition of human sodium-calcium exchanger NCX1 by XIP and SEA0400

Yan Zhao, yanli dong, Zhuoya Yu, Yue Li, Bo Huang, Qinru Bai, Yiwei Gao, Qihao Chen, Na Li, and Lingli He
DOI: 10.15252/emj.2023113785

Corresponding author(s): Yan Zhao (zhaoy@ibp.ac.cn)

Review Timeline:

Submission Date:	16th Feb 23
Editorial Decision:	21st Mar 23
Revision Received:	12th Jun 23
Editorial Decision:	1st Aug 23
Revision Received:	13th Oct 23
Editorial Decision:	3rd Nov 23
Revision Received:	10th Nov 23
Accepted:	15th Nov 23

Editor: William Teale

Transaction Report:

Dear Prof. Zhao,

Thank you again for the submission of your manuscript entitled "Mechanistic insight into allosteric inhibition of human sodium-calcium exchanger NCX1 by XIP and SEA0400" (EMBOJ-2023-113785). We have now received the reports from the referees, which I copy below.

As you can see from their comments, the referees agree that the work you present is potentially very interesting and important. That said, Referee 1 points to significant technical shortcomings which translate into the conclusions of the manuscript not being fully supported by the data as they are presented.

However, based on the overall interest expressed in the reports, I would like to invite you to address the comments of all referees in a revised version of the manuscript. I should add that it is The EMBO Journal policy to allow only a single major round of revision and that it is therefore important to resolve the main concerns at this stage. I believe the concerns of the referees are reasonable and addressable, but please contact me if you have any questions, need further input on the referee comments or if you anticipate any problems in addressing any of their points, I am always available over Zoom to help you through the revision process. Please, follow the instructions below when preparing your manuscript for resubmission.

I would also like to point out that as a matter of policy, competing manuscripts published during this period will not be taken into consideration in our assessment of the novelty presented by your study ("scooping" protection). We have extended this 'scooping protection policy' beyond the usual 3 month revision timeline to cover the period required for a full revision to address the essential experimental issues. Please contact me if you see a paper with related content published elsewhere to discuss the appropriate course of action.

Again, please contact me at any time during revision if you need any help or have further questions.

Thank you very much again for the opportunity to consider your work for publication. I look forward to your revision.

Best regards,

William

William Teale, Ph.D.
Editor
The EMBO Journal

When submitting your revised manuscript, please carefully review the instructions below and include the following items:

- 1) a .docx formatted version of the manuscript text (including legends for main figures, EV figures and tables). Please make sure that the changes are highlighted to be clearly visible.
- 2) individual production quality figure files as .eps, .tif, .jpg (one file per figure).
- 3) a .docx formatted letter INCLUDING the reviewers' reports and your detailed point-by-point response to their comments. As part of the EMBO Press transparent editorial process, the point-by-point response is part of the Review Process File (RPF), which will be published alongside your paper.
- 4) a complete author checklist, which you can download from our author guidelines ([https://wol-prod-cdn.literatumonline.com/pb-assets/embo-site/Author Checklist%20-%20EMBO%20J-1561436015657.xlsx](https://wol-prod-cdn.literatumonline.com/pb-assets/embo-site/Author%20Checklist%20-%20EMBO%20J-1561436015657.xlsx)). Please insert information in the checklist that is also reflected in the manuscript. The completed author checklist will also be part of the RPF.
- 5) Please note that all corresponding authors are required to supply an ORCID ID for their name upon submission of a revised manuscript.
- 6) We require a 'Data Availability' section after the Materials and Methods. Before submitting your revision, primary datasets produced in this study need to be deposited in an appropriate public database, and the accession numbers and database listed under 'Data Availability'. Please remember to provide a reviewer password if the datasets are not yet public (see

<https://www.embopress.org/page/journal/14602075/authorguide#datadeposition>). If no data deposition in external databases is needed for this paper, please then state in this section: This study includes no data deposited in external repositories. Note that the Data Availability Section is restricted to new primary data that are part of this study.

Note - All links should resolve to a page where the data can be accessed.

8) For data quantification: please specify the name of the statistical test used to generate error bars and P values, the number (n) of independent experiments (specify technical or biological replicates) underlying each data point and the test used to calculate p-values in each figure legend. The figure legends should contain a basic description of n, P and the test applied. Graphs must include a description of the bars and the error bars (s.d., s.e.m.).

9) We would also encourage you to include the source data for figure panels that show essential data. Numerical data can be provided as individual .xls or .csv files (including a tab describing the data). For 'blots' or microscopy, uncropped images should be submitted (using a zip archive or a single pdf per main figure if multiple images need to be supplied for one panel). Additional information on source data and instruction on how to label the files are available at .

10) We replaced Supplementary Information with Expanded View (EV) Figures and Tables that are collapsible/expandable online (see examples in <https://www.embopress.org/doi/10.15252/embj.201695874>). A maximum of 5 EV Figures can be typeset. EV Figures should be cited as 'Figure EV1, Figure EV2" etc. in the text and their respective legends should be included in the main text after the legends of regular figures.

12) Our journal encourages inclusion of *data citations in the reference list* to directly cite datasets that were re-used and obtained from public databases. Data citations in the article text are distinct from normal bibliographical citations and should directly link to the database records from which the data can be accessed. In the main text, data citations are formatted as follows: "Data ref: Smith et al, 2001" or "Data ref: NCBI Sequence Read Archive PRJNA342805, 2017". In the Reference list, data citations must be labeled with "[DATASET]". A data reference must provide the database name, accession number/identifiers and a resolvable link to the landing page from which the data can be accessed at the end of the reference. Further instructions are available at .

Additional instructions for preparing your revised manuscript:

We realize that it is difficult to revise to a specific deadline. In the interest of protecting the conceptual advance provided by the work, we recommend a revision within 3 months (19th Jun 2023). Please discuss the revision progress ahead of this time with the editor if you require more time to complete the revisions. Use the link below to submit your revision:

Referee #1:

The manuscript by Dong et al. describes structural studies of the human sodium/calcium exchanger NCX1.3. The work is incomplete. There are clear gaps in the presentation that undermine the confidence in the validity of the conclusions. The text is poorly written, contains numerous grammatical errors that confound understanding the authors' intent, and lacks a clear Discussion/summary of the main proposed findings.

The presented data raise questions about the quality of the sample and how such issues may impact the structural interpretation. Fig. S1D shows two prominent bands that are not NCX, 70 kD and ~50 kD. What are these?

Many loops in the structure are missing. The cartoon in 1D is misleading. Missing regions (5 from the text, 1-50, 384-404, 502-517, 638-653, 721-736) should be indicated.

Conservation in Fig. S3 is not well marked, making it very hard to follow whether elements are conserved, particularly with mjNCX, which was the model used to guide this structure. It appears that the conservation is very, very low with the archaeal exchanger mjNCX, which raises questions about how good a starting model, particularly with respect to helix register as the method state that the mjNCX model was to build the human NCX1.3. Further, given the apparent large differences in positions of multiple transmembrane elements of mjNCX and NCX1.3 (Fig.2B) one wonders how it was possible to use mjNCX as a guide. Notably, nowhere does the manuscript present a simple overlay of these two proteins.

Mutants designed to test the interactions with the inhibitory region, XIP are clearly affecting function (or expression) but the currents from the mutants are systematically smaller (and noisier) than wild-type with the IIQQ mutant being the worst. How close are these to background currents in the tested cells? That key control is missing.

The authors make claims about bound calcium ions but provide no evidence. Density to support the calcium ion assignment claims (line 140-145) is missing from the manuscript.

Density to support the SEA4000 binding is impossible to assess. The authors need to show a map of the entire region, not just some selected blobs in a tiny figure.

Overall the work may be an important start, but in its current state it can only be seen as incomplete.

Partial list of grammatical errors:

First line of the abstract 'inferences' cannot be the correct word. 'influences'?

Introduction line 35 'unitizes' Unclear what this word and sentence mean. 'utilizes'?

Referee #2:

Overall, this is an interesting paper describing the structure of the human Na²⁺/Ca²⁺ exchanger NCX1.3. While structures of bacterial homologues of this protein have been solved previously, the human protein has a large "regulatory loop" that is not present in the bacterial proteins. This so-called loop contains two calcium-binding domains (CBDs) as well as an exchanger inhibitory peptide (XIP). The authors show how the XIP is able to inactivate the exchange activity. Various small molecule inhibitors of the protein have also previously been developed as putative drugs. In solving the structure, they add one such an inhibitor to the protein. They show how this binds to the protein and suggest a mechanism of inhibition.

Though the nominal resolution of the data is relatively modest (3.5Å) and better for the transporter domain relative to the CBDs their interpretation of the structure is generally consistent with the resolution. The binding modes of the XIP and the inhibitor are supported by electrophysiology studies of various mutants.

The manuscript gives a good description of the structure and the associated patch-clamp data. However, they leave the paper hanging and don't make any conclusions. Thus, the reader has to make their own conclusions. They don't explicitly propose a mechanism of inactivation of the XIP, or discuss the role of the CBDs and calcium binding in the context of their structure.

The only real concern I have is how the density looks like for the SEA0400. The only evidence of the binding mode that they present is of density around the inhibitor. I would like to see a figure in the supplementary material showing the density of the inhibitor and the surrounding residues, otherwise it is difficult to ascertain the noise level in the maps. They mention that the inhibitor stabilises the protein. At present there is no evidence in the paper that the inhibitor even binds to the purified protein.

Minor points

Lines from 165-175 discuss creating models through the inverted repeats but the relevance of this is not explained.

Figure S8 suggests that TM2 changes conformation also in the inward-facing state. This figure doesn't seem to be referenced from the text and some explanation is needed to how it relates to their ideas.

It is difficult to see many of the structural features in Figure 1.

Sequence identity should be added in Figure S3. It is difficult to see how similar the sequences are.

It is difficult to see the "steric hinderance" in figure 2B

Line 197: K264 - to what - and 264 in what sequence?

Some proof reading is required - especially in the abstract.

Referee #3:

The manuscript by Dong et al. describes the structural characterization of the human Na/Ca exchanger NCX1.3 by cryo-EM in the presence of small molecule inhibitor SEA0400. The work is of high quality and presents interesting new findings about the architecture of this medically important transporter with a pivotal role in the excitability of cardiac myocytes. In particular, the organization and intramolecular interactions of the regulatory cytoplasmic Ca binding domains CBD1 and CBD2 and the XIP region is very insightful and aligns well with previously published functional characterization of these regions. Only a single structure in the inward open conformation (in complex with a small molecule antagonist) is included in the current work, presumably because high resolution structure determination under different conditions (eg apo) is not feasible due to instability of the purified protein in absence of stabilizing compounds or flexibility of the protein (as indicated by the authors). The manuscript shows a comparison to a structure of a non-human orthologue in the outward open state to draw conclusions about the putative conformational changes during the transport cycle of human NCX1.3. Furthermore, the manuscript includes Ca imaging with Fura-2 and electrophysiological data measuring the transport through NCX1.3 in reverse mode (Ca influx into cells in exchange for intracellular Na). These experiments confirm the functional activity of the construct used for purification structure determination and is utilized to characterize a range of mutants in regions of interest identified from the structure. The functional data nicely confirms the observations from the structure, such as the role of the XIP region for Na(in) -dependent inactivation of NCX1.3 and the loss off inhibition when mutating key residues involved in the binding the small molecule antagonist SEA0400. All in all, the findings are well supported by the experimental data and the authors present extensive validation of structural

findings by mutagenesis/functional studies. The experimental design is rigorous, and the quality of the EM data is sufficient for model building and for determining the binding site of the small molecule. An interesting insight is that the antagonist is not overlapping with the ion binding sites, suggesting an allosteric mode of action. This makes sense structurally because the compound interaction locks the transporter in an inward-open state and would explain insurmountable potency of this inhibitor under physiological ionic conditions. Overall, the work is well suited for publication in The EMBO Journal, but small revisions of the manuscript text and Figures would improve the readability of this manuscript.

Major points:

- Adding a movie file of a morph between the experimentally determined inward open state structure NCX1.3_IF (minus cytoplasmic domains) and the outward open state model NCX1.3_OF* (generated by comparison with the non-human orthologue) would make it easier to visualize the conformational changes that may happen during the transition between these states. The two states are compared in Figure 2, 4 and S5 but from the superpositions only the concerted movement of different regions is difficult to understand.
- It is a bit unclear why the transporter is only studied in reverse mode, since in the patch clamp experiments it should be possible to choose suitable ionic solutions in the pipette and perfusion for forward mode as well? Maybe this is obvious for people directly working with these transporters, but for readers who are mostly familiar with more distantly related transporters it would be helpful to explain why no forward mode measurements are included or if these have been attempted and whether any further mechanistic conclusions would be possible in forward mode.
- The manuscript is very short and no discussion is added after the results section. The first part of the result section is also written in a quite technical style and repeats some details which are already in the method section. A few topics would be interesting to read in the discussion, eg. a more detailed section about the therapeutic relevance of modifying NCX1.3 activity by an allosteric inhibitor like SEA0400 and whether the high selectivity of the compound over other SLC8 family members is desirable would be insightful. The difference in sensitivity to SEA0400 for NCX1-3 is briefly mentioned in lines 253-255, but it would make sense to explain the physiological consequences and the therapeutic relevance of these differences in the discussion. It is mentioned that SEA0400 preferentially inhibits the reverse mode (Ca²⁺ uptake)- is there a structural explanation why the inhibitor is more effectively antagonizing the reverse mode and can this be added to the discussion? Another topic that would be interesting for the discussion is whether any clinical variants are reported in any of the highlighted regions and whether there is potentially a rationale for the design of small molecule activators that target the observed inhibitory interdomain interactions which stabilize the protein in the inward open state and prohibit transition into the outward open state. These sections should be rewritten/added to the manuscript. In general, the manuscript suffers from small errors in grammar and the language can be improved.
- The main Figures have relatively small individual panels which makes it hard to see details described in the manuscript when the paper is printed as hard copy. In particular, Figure 4 B showing electron density around the small molecule SEA0400 is hard to see and it would make sense to either make it bigger, or alternatively add an extra panel with an enlarged inset or add a larger Figure to the extended Figure S2 which shows density maps for different protein regions, but not for the small molecule. In the current version, it is difficult to see how well the map supports the proposed binding mode of SEA0400 and the mutant data mostly confirms the binding site but is insufficient to confirm the exact pose of the molecule.

Minor points:

Line 17: inferences -> infers

Line 80: jaxtamembrane -> juxtamembrane

Table S1: The clash score is a bit high - can this be improved by further optimization of the model in extra refinement cycles?

Abstract, line 21 "stabilized at an inward-facing conformation" -> rephrase to "stabilized in an inward-facing conformation"

Main Figure 2-4 title: structure basis -> structural basis

The authors may want to consider a more diverse choice of titles for these three main Figures.

Figure 3 legend: descriptions for panels A-D in the legend do not match with the Figure which has panels A-E. The description of panel B is missing (or part of A?) and the other panel labels need to be shifted to match the Figure.

Figure 4 F: The color scheme is not ideal, because NCX1.3 is colored "in rainbow" and superposed with mjNCX in cyan, but the rainbow looks more like a blue/pink gradient, making it hard to see the details in this relatively small panel. The contrast is very faint too because of the semitransparent cartoon.

Figure S2 B: maps created by cryo-EM are typically not called "electron" density maps, this is used for crystallography, it is correct to call it instead "EM density maps" or "cryo-EM maps"

Figure S4 Title: Sequence and structural alignment-> this figure does not show any sequence alignments, only structural alignments.

Point-by-point response for

Mechanistic insight into allosteric inhibition of human sodium-calcium exchanger NCX1 by XIP and SEA0400

**Referee #1:**

The manuscript by Dong et al. describes structural studies of the human sodium/calcium exchanger NCX1.3. The
 work is incomplete. There are clear gaps in the presentation that undermine the confidence in the validity of the
 conclusions. The text is poorly written, contains numerous grammatical errors that confound understanding the
 authors' intent, and lacks a clear Discussion/summary of the main proposed findings.

**Reply:** We appreciate very much the reviewer's comment and his/her suggestions for improving our manuscript.

1. The presented data raise questions about the quality of the sample and how such issues may impact the
 structural interpretation. Fig. S1D shows two prominent bands that are not NCX, 70 kD and ~50 kD. What are
 these?

**Reply:** We thank the reviewer for pointing this out. The ~70 kD and ~50 kD protein bands observed in the
 Appendix Figure S1D (the original Figure S1D) SDS-PAGE gel are likely non-specific binding protein that were
 not completely removed during the purification process, rather than NCX protein samples. The 70 kD band may
 be a heat shock protein HSP70, which is produced as a molecular chaperone during protein expression.
 Although we added ATP and magnesium chloride during sample purification, these components were not
 entirely separated from the target protein. However, these non-specific proteins are not expected to impact our
 final structural analysis, as they represent only a small proportion of the total protein sample. Furthermore,
 during Cryo-EM data processing, we used 2D and 3D classification to specifically select and enrich the particles
 featuring transmembrane helices and discernible soluble domain of NCX1.3 and eliminated other junk particles.
 Therefore, we are confident that the non-specific binding proteins would not impact structure determination and
 our structural interpretation is reliable.

2. Many loops in the structure are missing. The cartoon in 1D is misleading. Missing regions (5 from the text, 1-
 50, 384-404, 502-517, 638-653, 721-736) should be indicated.

**Reply:** We thank the reviewer for this comment and have made the necessary revisions to Figure 1D. We have
 added the missing regions (1-50, 384-404, 502-517, 638-653, 721-736) to the cartoon and have represented them
 using dashed lines to clearly indicate the missing loops. The revised Figure 1D and the old Figure 1D are attached
 below for your convenience.

Figure1*. Revised and old versions of Figure 1D. **a.** The Revised version of Figure 1D. The missing regions in
 the NCX1.3 structure are shown in dashed lines which include residues 1-50, 384-404, 502-517, 638-653, and
 721-736. **b.** The old version of Figure 1D.

3. Conservation in Fig. S3 is not well marked, making it very hard to follow whether elements are conserved,
particularly with mjNCX, which was the model used to guide this structure. It appears that the conservation is very,
very low with the archaeal exchanger mjNCX, which raises questions about how good a starting model, particularly
with respect to helix register as the method state that the mjNCX model was to build the human NCX1.3. Further,
given the apparent large differences in positions of multiple transmembrane elements of mjNCX and NCX1.3
(Fig.2B) one wonders how it was possible to use mjNCX as a guide. Notably, nowhere does the manuscript present
a simple overlay of these two proteins.

**Reply:** Thank you for your comment. We have revised Appendix Figure S3 (the original Figure S3) to indicate
conserved residues and the sequence alignment was performed using Clustal^[1] and colored based on
conservation in Jalview^[2]. The revised Appendix Figure S3 is attached below for your convenience. Despite the
low conservation of residues between NCX and mjNCX, the ion-binding site exhibits a high level of conservation.
During the model building, we relied solely on the outward-facing conformation of the mjNCX model to provide a
rough estimation of the arrangement of certain transmembrane helices in the NCX structure. Specifically, we
focused on TM2-TM5 and TM7-TM10 since these regions in mjNCX aligned remarkably well with our map. The
Figures 2B and 4F demonstrated that TM2-TM5 and TM7-TM10 helices are superimposable between NCX and
the outward-facing mjNCX. Importantly, we did not rely on the mjNCX model to determine the helix register. Instead,
our map provided sufficient features in the transmembrane region (Appendix Figure S2E) that enabled us to
accurately determine the specific positions of each residue in the transmembrane helices, particularly with the help
of bulky amino acid residues. We have also modified the methods section, please see Pg 15 (Lines 500-507), it
reads now "The core domain (TM2-TM5 and TM7-TM10) of crystal structure of mjNCX (PDB ID: 3V5S) (Liao *et*
*al.*, 2012) was extracted using PyMOL software and then docked into the map as a rigid body using UCSF Chimera
(Pettersen *et al.*, 2004) to roughly determine the arrangement and connection of these transmembrane helices.
Each helix was further refined and adjusted based on the map using COOT. Additionally, the TM1 and TM6 helices
were built manually in COOT based on the density map. Finally, we conducted careful manual adjustments based
on the density of amino acid side chains, guided by well-resolved densities of residues with bulky side chains."

Appendix Figure S3. Full-length sequence alignment of NCX homologous. Related to Figure 2, Figure 3, and Figure 4.

The sequence alignment among human NCX1.3, human NCX2.1, human NCX3.1, human NCKX2 and *Methanococcus jannaschii* NCX. Sequence alignment was performed using Clustal and colored based on conservation in Jalview. Intensity of blue darkcyan indicates conservation at the position. Gaps are indicated by dashes. Secondary structural elements of hsNCX1.3 are indicated and marked above the sequence alignment. The XIP region is shaded in yellow. The exon B and exon D of NCX1.3 CBD2 are marked as red lines and purple lines, respectively. Na⁺/Ca²⁺ binding sites are highlighted in red. Residues important for SEA0400 binding are marked as green triangles.

[1] Larkin MA, Blackshields G, Brown NP, Chenna R, McGettigan PA, McWilliam H, Valentin F, Wallace IM, Wilm A, Lopez R, Thompson JD, Gibson TJ, Higgins DG, "Clustal W and Clustal X version 2.0" *Bioinformatics*, 23: 2947

(2007)]

[2] Waterhouse, A.M., Procter, J.B., Martin, D.M.A, Clamp, M., Barton, G.J (2009), "Jalview version 2: A Multiple
Sequence Alignment and Analysis Workbench," *Bioinformatics* **25** (9) 1189-1191 doi:
10.1093/bioinformatics/btp033

4. Mutants designed to test the interactions with the inhibitory region, XIP are clearly affecting function (or
expression) but the currents from the mutants are systematically smaller (and noisier) than wild-type with the IIQQ
mutant being the worst. How close are these to background currents in the tested cells? That key control is missing.

**Reply:** We appreciate reviewer's comment. In our study, we utilized whole-cell patch clamp techniques to measure
the currents of both wild-type (WT) and mutants under identical conditions, including the same membrane potential,
buffer conditions and temperature, etc. In this experimental setup, two primary factors that influence current
characteristics and amplitude are the properties of the protein itself and its expression level. While the expression
level primarily impacts the current amplitude, it does not alter the current kinetics. The reviewer pointed out that
the currents from mutants being smaller than the WT. We statistically analyzed the current amplitude of all of
recordings, but we did not find that there is significant difference between the WT and mutants in current amplitude
(Figure A). To explore the functional roles of the XIP region, previous reports^[1-3] have utilized the ratio of steady-
state current to peak current as an assessment of Na⁺-dependent inactivation. In our study, we adopted a similar
approach using the R10 value to represent current inactivation. The R10 value is determined by calculating the
ratio of the average current during the last 2 seconds in the presence of Ca²⁺ over a period of 10 seconds to the
peak current. These values provide an intuitive representation of the inactivation kinetics mediated by XIP, which
is independent of the magnitude of the peak current or the level of protein expression.

The reviewer also noted that the currents from mutants appeared to be noisier compared to the wild-type (WT).
Typically, noise in whole-cell patch clamp recordings primarily stems from the recording equipment, especially
when the experimental conditions are kept similar. All of electrophysiological experiments are performed on the
same equipment. We believe that the variation in noise levels might be misleading due to unclear data presentation.
In the original figure, we normalized the peak currents of both WT and mutants, resulting in different scale bars
and creating an illusion of varying noise levels. To address this concern, we have made modifications to Figure
3D by employing a unified scale bar. Our revised Figure, as depicted in Figure B, demonstrates that the WT and
mutants exhibit a comparable level of current noise.

Figure 2*. Whole-cell patch clamp recordings of NCX1.3. A. Summary data of peak current from NCX^{WT}, NCX1.3
101 ^{ΔXIP} and NCX1.3^{IIQQ}. Dots represent peak current for individual cell. Significance was determined by unpaired
Student's *t* test at a 95% confidence interval. Error bars indicate SEM. There is no statistical difference between
NCX^{WT} and NCX1.3^{ΔXIP} ($p = 0.6917$), NCX^{WT} and NCX1.3^{IIQQ} ($p = 0.5896$). B. Representative outward currents of

NCX1.3^{WT}, NCX1.3^{ΔXIP} and NCX1.3^{IQQ} were elicited by changing the extracellular Ca²⁺ from 0 to 1 mM for 10s.

[1] Chou A C, Ju Y T, Pan C Y. Calmodulin interacts with the sodium/calcium exchanger NCX1 to regulate
activity[J]. PloS one, 2015, 10(9): e0138856.

[2] Ottolia M, Nicoll D A, John S, et al. Interactions between Ca²⁺ binding domains of the Na⁺-Ca²⁺ exchanger
and secondary regulation[J]. Channels, 2010, 4(3): 159-162.

5. The authors make claims about bound calcium ions but provide no evidence. Density to support the calcium ion
assignment claims (line 140-145) is missing from the manuscript.

**Reply:** We appreciate reviewer's comment. Due to moderate resolution of our elucidated NCX1.3 map, we were
unable to observe the density of calcium ions. However, we speculate that CBD1 in our elucidated structure may
have bound to calcium ions based on two key aspects. Firstly, the CBD1 of NCX1 has four calcium ion binding
sites, and calcium ions exhibit a high affinity for CBD1 ($K_{d_{Ca1/Ca2}} \sim 10 \mu M$, $K_{d_{Ca3/Ca4}} \sim 0.4 \mu M$)^[1]. Considering that
we introduced a high concentration of Ca²⁺ (~1 mM) during the protein sample preparation, we believe that the
resolved CBD1 is likely to be associated with calcium ions. Secondly, previous studies on the CBD1-2 structure
have indicated that when CBD1 is bound to Ca²⁺, it adopts a tightly packed structure. Interestingly, in the
absence of Ca²⁺, the upper half of CBD1 unfolds, resulting in a loosely packed conformation^[2]. When comparing
the structure of CBD1 in NCX1.3 with the Ca²⁺-bound CBD1 in NCX1^[3], an RMSD of approximately 0.9 Å for 116
Cα-pairs was observed (Figure EV1A). Therefore, based on these factors, we speculate that CBD1 is likely to be
bound to calcium ions in our elucidated structure.

In the revised manuscript, please see Pg 5 (lines 138-148), we adjusted our related discussion. It now reads
"The CBD1 and CBD2 are organized in a head-to-tail tandem through a short linker, generating a ~126° angle
between CBD1 and CBD2 domains (Figures 1A and EV1). The CBD1 domain of the NCX1 protein contains four
calcium ion binding sites, with Ca²⁺ ions displaying a strong affinity for CBD1 ($K_{d_{Ca1/Ca2}} \sim 10 \mu M$, $K_{d_{Ca3/Ca4}} \sim 0.4$
μM) (Hilge *et al.*, 2009a). Moreover, Previous studies suggested that the upper half of isolated CBD1 loses its
structural integrity in the absence of Ca²⁺ (Hilge *et al.*, 2006). In contrast, the CBD1 domain in our NCX1.3
structure exhibits a well-defined conformation, with an RMSD of approximately 0.9 Å for 116 Cα-pairs when
compared to the isolated Ca²⁺-bound CBD1 (Figure EV1A). Considering these findings, along with the fact that
we added approximately 1 mM Ca²⁺ during the sample preparation, we speculate that Ca²⁺ ions likely bind to the
CBD1 domain in our elucidated NCX1.3 structure. However, due to the moderate resolution of our resolved
NCX1.3 map, we were unable to observe the density corresponding to Ca²⁺ ions.

[1] Hilge M, Aelen J, Fource A, et al. Ca²⁺ regulation in the Na⁺/Ca²⁺ exchanger features a dual electrostatic
switch mechanism[J]. Proceedings of the National Academy of Sciences, 2009, 106(34): 14333-14338.

[2] Hilge M, Aelen J, Vuister G W. Ca²⁺ regulation in the Na⁺/Ca²⁺ exchanger involves two markedly different
Ca²⁺ sensors[J]. Molecular cell, 2006, 22(1): 15-25.

[3] Nicoll D A, Sawaya M R, Kwon S, et al. The crystal structure of the primary Ca²⁺ sensor of the Na⁺/Ca²⁺
exchanger reveals a novel Ca²⁺ binding motif[J]. Journal of Biological Chemistry, 2006, 281(31): 21577-
21581.

6. Density to support the SEA4000 binding is impossible to assess. The authors need to show a map of the entire
region, not just some selected blobs in a tiny figure.

**Reply:** We appreciate this comment. We have taken your suggestion and added a new panel in Appendix Figure
S2 (the original Figure S2) to show the map of the entire region around the SEA0400 binding site from two different
views. The density around the SEA0400 binding site is well-defined and the surrounding amino acid residues,
including W210, F248, and V134, have clear side chain densities, confirming the reliability of the SEA0400 binding.
The Appendix Figure S2E is attached below for your convenience.

Figure 3*. The added Appendix Figure S2E. Density of residues and SEA0400 are shown as mesh.

Overall the work may be an important start, but in its current state it can only be seen as incomplete.

Reply: We thank the reviewer for his/her suggestion. We have improved our manuscript according to the reviewer's suggestion, including correcting grammatical errors and adding the discussion section in the revised manuscript.

Partial list of grammatical errors:

1. First line of the abstract 'inferences' cannot be the correct word. 'influences'?

Reply: We thank the reviewer for this comment. This typo has been corrected in the revised version.

2. Introduction line 35 'unitizes' Unclear what this word and sentence mean. 'utilizes'?

Reply: We thank the reviewer for pointing out this typo and we have corrected it in our revised manuscript.

Referee #2:

Overall, this is an interesting paper describing the structure of the human $\text{Na}^{2+}/\text{Ca}^{2+}$ exchanger NCX1.3. While structures of bacterial homologues of this protein have been solved previously, the human protein has a large "regulatory loop" that is not present in the bacterial proteins. This so-called loop contains two calcium-binding domains (CBDs) as well as an exchanger inhibitory peptide (XIP). The authors show how the XIP is able to inactivate the exchange activity. Various small molecule inhibitors of the protein have also previously been developed as putative drugs. In solving the structure, they add one such an inhibitor to the protein. They show how this binds to the protein and suggest a mechanism of inhibition.

Though the nominal resolution of the data is relatively modest (3.5\AA) and better for the transporter domain relative to the CBDs their interpretation of the structure is generally consistent with the resolution. The binding modes of the XIP and the inhibitor are supported by electrophysiology studies of various mutants.

Reply: We appreciate the reviewer's positive comments very much.

1. The manuscript gives a good description of the structure and the associated patch-clamp data. However, they leave the paper hanging and don't make any conclusions. Thus, the reader has to make their own conclusions. They don't explicitly propose a mechanism of inactivation of the XIP, or discuss the role of the CBDs and calcium binding in the context of their structure.

Reply: We appreciate the reviewer's comments. We have discussed the functional roles of XIP and CBDs in our revised manuscript. In particular, we believe that the XIP region plays an important role in underlying the preferential inhibition of NCX1.3 at reverse mode by SEA0400.

In the revised manuscript, please see Pgs 10-12 (lines 302-367). We have attached the discussion section here for your convenience. It now reads "The sodium-calcium exchanger (NCX) can mediate the exchange of Na^{+} and

181 Ca^{2+} in both forward and reverse transport modes, regulating the Ca^{2+} homeostasis in cells and participating in
various physiological and pathological processes. In this study, we elucidated the cryo-electron microscopy (cryo-
EM) structure of the inward-facing conformation of human NCX1.3 in the presence of the SEA0400 specific
inhibitor. The XIP region, responsible for mediating Na^+ -dependent inactivation, is positioned within a groove
situated between TMD and CBD2, stabilizing the inward-facing conformation and preventing a transition to the
outward-facing state by clashing with gating helices TMs^{1/6} at the outward-facing state (Figure 3C). However, due
to limited resolution, the precise localization of Na^+ within the structure could not be determined, preventing the
elucidation of its molecular mechanism in facilitating XIP binding. Cytoplasmic Ca^{2+} acts not only as a substrate
for the $\text{Na}^+/\text{Ca}^{2+}$ exchanger, but also regulates its activity of NCX by binding to the intracellular domains, CBD1
and CBD2. CBD1 contains four Ca^{2+} binding sites ($K_{d_{\text{Ca}1/\text{Ca}2}}: \sim 10 \mu\text{M}$, $K_{d_{\text{Ca}3/\text{Ca}4}}: \sim 0.4 \mu\text{M}$) (Hilge *et al.*, 2009a).
The number of Ca^{2+} binding sites in CBD2 is determined by its mutually exclusive exons, exon A and exon B. The
CBD2 with exon A has two Ca^{2+} binding sites ($K_{d_{\text{Ca}1}}: 9 \mu\text{M}$, $K_{d_{\text{Ca}2}}: 1 \mu\text{M}$) (Hilge *et al.*, 2009a), whereas CBD2 with
exon B lacks Ca^{2+} binding sites (Hilge *et al.*, 2009a). The NCX1.3 contains exon B and thus we speculate that only
CBD1 may bind with Ca^{2+} in current structure. Previous investigations showed that binding of Ca^{2+} to CBD1
activates the transport activity of the NCX protein (Khananshvili, 2020) and the presence of Ca^{2+} leads to the
stabilization of a specific conformation of CBD1 and CBD2 (Giladi *et al.*, 2012). However, when Ca^{2+} dissociates,
CBD1 loses its structural integrity (Hilge *et al.*, 2006). The association of Ca^{2+} with CBD1 has been also reported
to have varying effects on the conformation of CBD2 in different splicing isoforms, thereby exerting distinct
modulatory effects on the transport activity (Khananshvili, 2020). Nevertheless, we have been unable to reach a
conclusive understanding of how Ca^{2+} bound to CBD1 activates the transporter allosterically, primarily due to the
limitation of having only one structure that is potentially bound with Ca^{2+} . Further investigations and additional
structures are needed to shed more light on this matter. Despite CBD2 of NCX1.3 being unable to bind with Ca^{2+} ,
structural analysis reveals that the potential Ca^{2+} binding site of CBD2 with exon A is located in close proximity to
the XIP binding site (Figure 3C). In the structure of NCX1.3, CBD2 exhibits a negatively charged electrostatic
surface, which facilitates the binding of the positively charged XIP region with the TMD (Figure 3C). We further
hypothesize that the binding of Ca^{2+} to CBD2 with exon A would neutralize this negative electrostatic potential,
consequently reducing the binding affinity of the XIP region. This, in turn, could result in a reduction or even
complete elimination of XIP-mediated inactivation.

The selective inhibitor SEA0400 predominantly blocks NCX1 and exerts almost no influence on NCX3 (Iwamoto
*et al.*, 2004b). In our structure, it binds to an allosteric binding site where most of the residues are conserved,
except for F248, which is substituted with L in NCX3 (Figure 4C). Functional experiments further confirm the crucial
role of this residue in subtype selectivity (Figure 4E), which provides a solid foundation for the rational design of
subtype-specific drugs. Previous investigations have demonstrated that the SEA0400 is preferable for inhibiting
the reverse mode of NCX (Tanaka, Nishimaru *et al.*, 2002), which is responsible for the uptake of extracellular
Ca^{2+} . However, the molecular basis for its selectivity towards the reverse mode over the forward mode remains
unknown. Understanding this molecular basis is crucial for rational development of pharmacological inhibitors that
specifically target NCX1 in the reverse mode, holding promising therapeutic potential in preventing Ca^{2+} overload
and electrical dysfunction in the context of ischemia/reperfusion injury and heart failure (Chen & Li, 2012a;
Imahashi *et al.*, 2005). When the NCX operates in the reverse mode, the high concentration of intracellular Na^+
plays a dual role. On one hand, it is required to act as a driven ion, while on the other hand, it facilitates the
inactivation of NCX1.3 through the XIP region, thereby effectively preventing cells from Ca^{2+} overloading by NCX
during the pathological Na^+ accumulation (He *et al.*, 2000; Hilgemann *et al.*, 1992; Li *et al.*, 1991; Matsuoka *et al.*,
1993). As the sample is purified in the presence of high concentration Na^+ , we speculate that the complex structure
may represent an inactive state, primarily associated with the reverse mode. The XIP engages with the TM1-2
loop and plays a crucial role in the inactivation process by stabilizing the inward-facing conformation and
preventing a transition to the outward-facing state. Furthermore, these interactions between the XIP and TM1-2
loop also maintain a specific conformation of TM2ab, which is essential for the allosteric binding of SEA0400. Our
calcium imaging experiments have also demonstrated that the XIP is vital for the sensitivity of NCX to SEA0400

(Figures 4E and EV4F). In the forward mode, we speculate that the stability of the SEA0400 binding pocket is
compromised compared to the reverse mode, because the XIP may not effectively interact with the TM1-2 loop at
lower intracellular Na⁺, supported by the minimal inactivation observed in the forward mode over time (Iwamoto *et*
*al.*, 2004a, Matsuoka, Nicoll *et al.*, 1995), thereby failing to stabilize the SEA0400 binding pocket. Hence, we
hypothesize that the Na⁺-dependent XIP-mediated inactivation, which is more prominently observed in the reverse
mode, leads to a more stable SEA0400 binding pocket compared to the forward mode, resulting in the selective
inhibition of the reverse mode by the SEA0400.”.

2. The only real concern I have is how the density looks like for the SEA0400. The only evidence of the binding
mode that they present is of density around the inhibitor. I would like to see a figure in the supplementary material
showing the density of the inhibitor and the surrounding residues, otherwise it is difficult to ascertain the noise
level in the maps. They mention that the inhibitor stabilises the protein. At present there is no evidence in the paper
that the inhibitor even binds to the purified protein.

**Reply:** We appreciate your valuable comment. We have taken your suggestion and added a new panel in
Appendix Figure S2 (the original Figure S2) to show the map of the entire region surrounding the SEA0400
molecule from two different views. In this updated Figure, the density corresponding to the SEA0400 molecule
and the surrounding amino acid residues are well defined. Notably, the side chains of residues W210, F248, and
V134 have clear densities, further confirming the authenticity of the SEA0400 molecule and ensuring that it is not
a result of noise. The Appendix Figure S2E is attached below for your convenience.

Figure 4*. The added Appendix Figure S2E. Density of residues and SEA0400 are shown as mesh.

Additionally, we also deeply appreciate the insightful comment from the reviewer regarding the stabilization of
NCX1.3 by SEA0400. We fully agree with the reviewer that it is crucial to provide evidence supporting the binding
and stabilizing properties of SEA0400. To evaluate the effect of SEA0400 on stability of NCX1.3, we conducted
Thermofluor assay, which is a well-established and widely used experimental method for measuring protein
thermal stability. It turns out that the melting temperature is enhanced by ~10°C in the presence of the SEA0400,
demonstrating the SEA0400 indeed binds to and stabilizes the purified NCX1.3 sample. We have included this
result as a new panel in Appendix Figure S1 (the original Figure S1), which is attached here for your convenience
(Figure 5*).

We have also revised related discussion in our manuscript, please see Pg 3 (lines 96-98), it now reads " Meanwhile,
the thermofluor assay demonstrated that the NCX1.3-selective inhibitor SEA0400 can raise melting temperature
of NCX1.3 by ~10°C (Appendix Figure S1E). Therefore, during protein expression, purification, and grid
preparation, we included the SEA0400 to improve the stability and conformational homogeneity of the NCX1.3
sample."

We have also included a method section for the thermo stability assay. Please see Pg 13 (lines 411-428), it now
reads "To evaluate the stability of the purified NCX1.3 transporter in the presence or absence of SEA0400, thermal
stability assays were used as described previously (Alexandrov *et al.* 2008). Thermofluor utilizes the binding of a
dye called N-[4-(7-diethylamino-4-methyl-3-coumarinyl) phenyl] maleimide (CPM; Invitrogen) which reacts with
thiol groups of cysteine residues by maleimide reaction chemistry. The purified NCX1.3 protein sample with a

concentration of 6 mg/mL was finally diluted to a final concentration of 0.2 mg/mL in a total volume of the 25 μ L
assay buffer (20 mM HEPES, pH 7.5, 150 mM NaCl, 0.007% (w/v) GDN). Freshly made CPM was added to the
assay buffer by diluting the stock (4 mg/mL in DMSO) 16-fold, resulting in a solution of 1 μ L of CPM per 25 μ L
PCR tube. For the thermal stability assays investigating the NCX1.3 protein in the presence of SEA0400, the
reaction mixture was supplemented with 100 μ M SEA0400 and further incubated for 15 min on ice in the dark.
Thermal denaturation was performed using a RotorGene 6600 (Qiagen, USA) real-time PCR instrument with a
ramp rate of 1.0°C/min and a step-wise temperature increase from 25 to 95°C in 1.0°C/cycle increments (ramp
rate of 1.0°C/min) and with an equilibration time of 5 s at each temperature. The fluorescence signal was measured
using filters with an excitation wavelength of 384 nm and an emission wavelength of 470 nm. Data were exported
for analysis in Microsoft Excel 2016 and analyzed with GraphPad Prism 9 software (GraphPad software).”

Figure 5*. The added Appendix Figure S1E. Thermal stability assay as described in the methods section. The
addition of 100 μ M SEA0400 increased the thermal stability of NCX1.3.

Minor points

1. Lines from 165-175 discuss creating models through the inverted repeats but the relevance of this is not
explained.

**Reply:** Thank you for your valuable comment. In our study, we successfully obtained the inward-facing
conformational structure of NCX1.3. To investigate the conformational changes occurring during the ion
exchange process, we initially compared the NCX1.3 structure with the outward-facing conformation of the
prokaryotic mjNCX. However, considering the relatively low sequence identity between mjNCX and NCX1.3, we
had concerns regarding the potential limitations of this comparison. To address this, we employed a common
approach by creating a model of the outward-facing conformation based on internal symmetry of NCX1.3^[1-3]. We
then also compared this model with the inward-facing conformation of NCX1.3. Both structural comparisons
revealed that the sliding movement of the TM1/6 helices plays a critical role in determining the transition
between the inward- and outward-facing conformations.

[1] Liao J, Li H, Zeng W, et al. Structural insight into the ion-exchange mechanism of the sodium/calcium
exchanger[J]. Science, 2012, 335(6069): 686-690.

[2] Forrest L R, Zhang Y W, Jacobs M T, et al. Mechanism for alternating access in neurotransmitter
transporters[J]. Proceedings of the National Academy of Sciences, 2008, 105(30): 10338-10343.

[3] Hu N J, Iwata S, Cameron A D, et al. Crystal structure of a bacterial homologue of the bile acid sodium
symporter ASBT[J]. Nature, 2011, 478(7369): 408-411.

2. Figure S8 suggests that TM2 changes conformation also in the inward-facing state. This figure doesn't seem to
be referenced from the text and some explanation is needed to how it relates to their ideas.

**Reply:** We appreciate reviewer's comment. In Figure EV5 (the original Figure S8), we presented a comparison
of the inward-facing states of NCX1.3, AfCAX, Vcx1, and YfkE. This Figure, on the one hand, revealed the
presence of conformational heterogeneity in TM2ab among these structures, on the other hand, it showed that
the TM2 is less bended in NCX1.3 compared to the other structures. This reduced bending can be attributed to

the interaction between XIP and the loop1-2 region that connects to TM2ab (Fig EV3B), which is not observed in
the other structures. Consequently, this interaction indirectly stabilizes TM2ab in its current position, thereby
preventing further dilation. Interestingly, the current position of TM2ab is crucial for the binding of SEA0400, as
SEA0400 forms extensive interactions with TM2ab (Fig EV1D). Our functional experiments (Figs 4 and EV4)
demonstrated that in the absence of XIP, the inhibitory effects of SEA0400 were also weakened, suggesting that
the XIP region reduces conformational heterogeneity of TM2ab of NCX1.3^{IF} and stabilizes it at a certain
conformation that preferentially binds with SEA0400, and thus facilitate the inhibitory function of SEA0400.

To better correlate the Figure EV5B and our conclusion, we revised our manuscript. Please see Pg 9 (lines 269-
276), it now reads: "Structural comparison of previously determined inward-facing CAX homologues (Nishizawa
*et al.*, 2013; Waight *et al.*, 2013; Wu *et al.*, 2013b) revealed that TM2ab exhibits substantial conformational
heterogeneity (Figure EV5B). Among these conformations, TM2 showed less bending in NCX1.3, likely due to
the extensive interactions between Loop1-2 and the XIP element that indirectly stabilize TM2ab in its current
position (Fig EV2B). Importantly, this stabilization is critical for SEA0400 binding, as SEA0400 molecule forms
extensive interactions with TM2ab (Fig EV1D). We speculate that the enhanced sensitivity of SEA0400 to NCX1
via XIP is due to the fact that XIP stabilizes TM2ab of NCX1.3^{IF} at a certain conformation that preferentially binds
with SEA0400, and thus the XIP and SEA0400 synergistically inhibit Ca²⁺-uptake activity of NCX1."

3. It is difficult to see many of the structural features in Figure 1.

**Reply:** Thank you for your comment. We have added an additional view that better illustrates the overall structure of
NCX, including the relative spatial arrangement of the TMD and CBD domains, as well as the arrangement of
the TMs and the location of the N- and C-termini, along with the position of XIP. Additionally, we have added some
labels to the Figure to highlight key features. The revised Figure 1 is attached below for your convenience.

4. Sequence identity should be added in Figure S3. It is difficult to see how similar the sequences are.

**Reply:** Thanks for your kind reminder. We have updated Appendix Figure S3 (the original Figure S3) to show the
sequence identity and similarity, and the sequence alignment was performed using Clustal^[1] and colored based
on conservation in Jalview^[2]. The revised Appendix Figure S3 is attached below for your convenience.

Appendix Figure S3. Full-length sequence alignment of NCX homologous. Related to Figure 2, Figure 3, and Figure 4.

The sequence alignment among human NCX1.3, human NCX2.1, human NCX3.1, human NCKX2 and *Methanococcus jannaschii* NCX. Sequence alignment was performed using Clustal and colored based on conservation in Jalview. Intensity of blue darkcyan indicates conservation at the position. Gaps are indicated by dashes. Secondary structural elements of hsNCX1.3 are indicated and marked above the sequence alignment. The XIP region is shaded in yellow. The exon B and exon D of NCX1.3 CBD2 are marked as red lines and purple lines, respectively. Na⁺/Ca²⁺ binding sites are highlighted in red. Residues important for SEA0400 binding are marked as green triangles.

[1] Larkin MA, Blackshields G, Brown NP, Chenna R, McGettigan PA, McWilliam H, Valentin F, Wallace IM, Wilm A, Lopez R, Thompson JD, Gibson TJ, Higgins DG, "Clustal W and Clustal X version 2.0" Bioinformatics, 23:

2947 (2007)]
[2] Waterhouse, A.M., Procter, J.B., Martin, D.M.A, Clamp, M., Barton, G.J (2009), "Jalview version 2: A
Multiple Sequence Alignment and Analysis Workbench," *Bioinformatics* **25** (9) 1189-1191 doi:
10.1093/bioinformatics/btp033

5. It is difficult to see the "steric hinderance" in figure 2B

**Reply:** Thank you for your comment. We have made changes to Figure 2 by adding a superimposed surface
representation of XIP. The revised figure clearly demonstrates the steric hindrance between the TM1 and TM6
helices of mjNCX and the XIP surface. In particular, the C-terminal end of TM1 is now shown to be almost entirely
obscured by the XIP surface. The revised Figure 2 is attached below for your convenience.

6. Line 197: K264 - to what - and 264 in what sequence?

**Reply:** We thank the reviewer for pointing this out. In previous studies, the mutation K229Q of canine NCX1.1
abolished [Na⁺]_{in}-dependent inactivation, which corresponds to K264 in human NCX1.3^[1]. We have changed this
in the revised manuscript. Please see Pg 7 (lines 200-202). It now reads "A previous report indicates that the point
mutation of K229Q of canine NCX1.1 abolishes [Na⁺]_{in}-dependent inactivation of the NCX, which corresponds to
K264 in human NCX1.3 (Matsuoka *et al*, 1997b).".

[1] Matsuoka S, Nicoll D A, He Z, et al. Regulation of the cardiac Na⁺-Ca²⁺ exchanger by the endogenous XIP
region[J]. *The Journal of general physiology*, 1997, 109(2): 273-286.

Some proof reading is required - especially in the abstract.

**Reply:** We thank the reviewer for pointing this out. we have thoroughly proofread the manuscript and made
necessary changes. The revised version of the manuscript highlights all the changes made.

Referee #3:

The manuscript by Dong et al. describes the structural characterization of the human Na/Ca exchanger NCX1.3
by cryo-EM in the presence of small molecule inhibitor SEA0400. The work is of high quality and presents
interesting new findings about the architecture of this medically important transporter with a pivotal role in the
excitability of cardiac myocytes. In particular, the organization and intramolecular interactions of the regulatory
cytoplasmic Ca binding domains CBD1 and CBD2 and the XIP region is very insightful and aligns well with
previously published functional characterization of these regions. Only a single structure in the inward open
conformation (in complex with a small molecule antagonist) is included in the current work, presumably because
high resolution structure determination under different conditions (eg apo) is not feasible due to instability of the
purified protein in absence of stabilizing compounds or flexibility of the protein (as indicated by the authors). The
manuscript shows a comparison to a structure of a non-human orthologue in the outward open state to draw
conclusions about the putative conformational changes during the transport cycle of human NCX1.3. Furthermore,
the manuscript includes Ca imaging with Fura-2 and electrophysiological data measuring the transport through
NCX1.3 in reverse mode (Ca influx into cells in exchange for intracellular Na). These experiments confirm the

functional activity of the construct used for purification structure determination and is utilized to characterize a
range of mutants in regions of interest identified from the structure. The functional data nicely confirms the
observations from the structure, such as the role of the XIP region for Na(in) -dependent inactivation of NCX1.3
and the loss of inhibition when mutating key residues involved in the binding the small molecule antagonist
SEA0400. All in all, the findings are well supported by the experimental data and the authors present extensive
validation of structural findings by mutagenesis/functional studies. The experimental design is rigorous, and the
quality of the EM data is sufficient for model building and for determining the binding site of the small molecule.
An interesting insight is that the antagonist is not overlapping with the ion binding sites, suggesting an allosteric
mode of action. This makes sense structurally because the compound interaction locks the transporter in an
inward-open state and would explain insurmountable potency of this inhibitor under physiological ionic conditions.
Overall, the work is well suited for publication in The EMBO Journal, but small revisions of the manuscript text and
Figures would improve the readability of this manuscript.

**Reply:** We appreciate very much the reviewer's comment and his/her suggestions for improving our manuscript.

Major points:

1. Adding a movie file of a morph between the experimentally determined inward open state structure NCX1.3_IF
(minus cytoplasmic domains) and the outward open state model NCX1.3_OF* (generated by comparison with the
non-human orthologue) would make it easier to visualize the conformational changes that may happen during the
transition between these states. The two states are compared in Figure 2, 4 and S5 but from the superpositions
only the concerted movement of different regions is difficult to understand.

**Reply:** Thank you very much for your valuable suggestion. We have created a movie file that demonstrates the
morph between the experimentally determined inward open state structure NCX1.3^{IF} and the outward open state
model NCX1.3^{OF*}. We have included this movie as "Movie EV1" in the Supplementary Materials section, and we
reference it in line 178 of the manuscript. The legend for the movie is as follows:

The N-domain and C-domain of NCX (Sodium-Calcium Exchanger) are represented by red and blue cartoons,
respectively, while the ion binding sites are shown as sticks. XIP (Exchanger Inhibitory Peptide) is depicted in
yellow cartoon, and SEA0400 is represented by green sticks. SEA0400 and XIP lock NCX1.3 in the inward-facing
conformation. Upon dissociation of SEA0400 and XIP from their respective positions, TMs1/6 of NCX1.3 undergo
vertical sliding along the membrane plane, accompanied by conformational changes in TM2a/TM7a. As a result,
the ion binding sites alternate exposure between the intracellular and extracellular sides, allowing for
conformational switching between inward-facing and outward-facing states.

You can view the movie using the following link: <https://drive.google.com/file/d/1Imk-848e7FgydthB1Li3Th1MkCmC-tT-/view?usp=sharing>

2. It is a bit unclear why the transporter is only studied in reverse mode, since in the patch clamp experiments it
should be possible to choose suitable ionic solutions in the pipette and perfusion for forward mode as well? Maybe
this is obvious for people directly working with these transporters, but for readers who are mostly familiar with
more distantly related transporters it would be helpful to explain why no forward mode measurements are included
or if these have been attempted and whether any further mechanistic conclusions would be possible in forward
mode.

**Reply:** We appreciate reviewer's suggestion. The NCX can operate in both forward and reverse modes depending
on the electrochemical gradients of Na⁺ and Ca²⁺ across the plasma membrane, as well as the membrane potential.
Specifically, when the pipette solution contained high Na⁺ and low concentration Ca²⁺ (1 μM derived from 5 mM
EGTA plus 4.28 mM CaCl₂), and the perfusion switch from 0 mM to 1 mM Ca²⁺-containing bath solution, it enabled
recording of outward current in the reverse mode. To record inward current in the forward mode, the perfusion
was switched from a Na⁺-free solution to a high Na⁺ solution, while the intracellular concentration of Ca²⁺ was
relatively high, reaching millimolar level. In our study, we performed whole-cell clamp patch experiments in the

reverse mode to investigate the functional roles of the XIP region in Na⁺-dependent inactivation, a commonly employed approach in previous studies exploring the functional roles of XIP^[1-2]. Thus, we believe that the evaluation of the XIP's functional role in deactivation kinetics in our study is adequately supported by the recording data obtained in the reverse mode. Additionally, we did perform the forward mode measurement to verify the construct is functional. However, the currents recorded from a single cell multiple times appear to be inconsistent, which was likely due to the cytotoxicity and seal instability caused by the high concentration of Ca²⁺ in the pipette. Consequently, drawing meaningful conclusions from the forward mode using our experimental setup proved to be challenging.

[1] Matsuoka S, Nicoll DA, Reilly RF, Hilgemann DW, Philipson KD. Initial localization of regulatory regions of the cardiac sarcolemmal Na⁺-Ca²⁺ exchanger. *Proc Natl Acad Sci U S A*. 1993 May 1;90(9):3870-4.

[2] Matsuoka S, Nicoll DA, He Z, Philipson KD. Regulation of cardiac Na⁺-Ca²⁺ exchanger by the endogenous XIP region. *J Gen Physiol*. 1997 Feb;109(2):273-86.

3. The manuscript is very short and no discussion is added after the results section. The first part of the result section is also written in a quite technical style and repeats some details which are already in the method section. A few topics would be interesting to read in the discussion, eg. a more detailed section about the therapeutic relevance of modifying NCX1.3 activity by an allosteric inhibitor like SEA0400 and whether the high selectivity of the compound over other SLC8 family members is desirable would be insightful. The difference in sensitivity to SEA0400 for NCX1-3 is briefly mentioned in lines 253-255, but it would make sense to explain the physiological consequences and the therapeutic relevance of these differences in the discussion. It is mentioned that SEA0400 preferentially inhibits the reverse mode (Ca²⁺ uptake)- is there a structural explanation why the inhibitor is more effectively antagonizing the reverse mode and can this be added to the discussion? Another topic that would be interesting for the discussion is whether any clinical variants are reported in any of the highlighted regions and whether there is potentially a rationale for the design of small molecule activators that target the observed inhibitory interdomain interactions which stabilize the protein in the inward open state and prohibit transition into the outward open state. These sections should be rewritten/added to the manuscript. In general, the manuscript suffers from small errors in grammar and the language can be improved.

Reply: Thanks for the reviewer's suggestions. We have added a discussion on the function of XIP and the regulation of CBD, as well as the selectivity of SEA0400 in the revised manuscript. Additionally, we have also corrected some grammar errors based on the reviewer's suggestions.

Please see Pgs 10-12 (lines 301-367). It now reads "The sodium-calcium exchanger (NCX) can mediate the exchange of Na⁺ and Ca²⁺ in both forward and reverse transport modes, regulating the Ca²⁺ homeostasis in cells and participating in various physiological and pathological processes. In this study, we elucidated the cryo-electron microscopy (cryo-EM) structure of the inward-facing conformation of human NCX1.3 in the presence of the SEA0400 specific inhibitor. The XIP region, responsible for mediating Na⁺-dependent inactivation, is positioned within a groove situated between TMD and CBD2, stabilizing the inward-facing conformation and preventing a transition to the outward-facing state by clashing with gating helices TMs^{1/6} at the outward-facing state (Fig 3C). However, due to limited resolution, the precise localization of Na⁺ within the structure could not be determined, preventing the elucidation of its molecular mechanism in facilitating XIP binding. Cytoplasmic Ca²⁺ acts not only as a substrate for the Na⁺/Ca²⁺ exchanger, but also regulates its activity of NCX by binding to the intracellular domains, CBD1 and CBD2. CBD1 contains four Ca²⁺ binding sites (Kd_{Ca1/Ca2}: ~10 μM, Kd_{Ca3/Ca4}: ~0.4 μM) (Hilge et al., 2009a). The number of Ca²⁺ binding sites in CBD2 is determined by its mutually exclusive exons, exon A and exon B. The CBD2 with exon A has two Ca²⁺ binding sites (Kd_{Ca1}: 9 μM, Kd_{Ca2}: 1 μM) (Hilge et al., 2009a), whereas CBD2 with exon B lacks Ca²⁺ binding sites (Hilge et al., 2009a). The NCX1.3 contains exon B and thus we speculate that only CBD1 may bind with Ca²⁺ in current structure. Previous investigations showed that binding of Ca²⁺ to CBD1 activates the transport activity of the NCX protein (Khananshvili, 2020) and the presence of Ca²⁺ leads to the stabilization of a specific conformation of CBD1 and CBD2 (Giladi et al., 2012). However, when Ca²⁺

dissociates, CBD1 loses its structural integrity (Hilge et al., 2006). The association of Ca^{2+} with CBD1 has been
also reported to have varying effects on the conformation of CBD2 in different splicing isoforms, thereby exerting
distinct modulatory effects on the transport activity (Khananshvili, 2020). Nevertheless, we have been unable to
reach a conclusive understanding of how Ca^{2+} bound to CBD1 activates the transporter allosterically, primarily
due to the limitation of having only one structure that is potentially bound with Ca^{2+} . Further investigations and
additional structures are needed to shed more light on this matter. Despite CBD2 of NCX1.3 being unable to bind
with Ca^{2+} , structural analysis reveals that the potential Ca^{2+} binding site of CBD2 with exon A is located in close
proximity to the XIP binding site (Fig 3C). In the structure of NCX1.3, CBD2 exhibits a negatively charged
electrostatic surface, which facilitates the binding of the positively charged XIP region with the TMD (Fig 3C). We
further hypothesize that the binding of Ca^{2+} to CBD2 with exon A would neutralize this negative electrostatic
potential, consequently reducing the binding affinity of the XIP region. This, in turn, could result in a reduction or
even complete elimination of XIP-mediated inactivation.

The selective inhibitor SEA0400 predominantly blocks NCX1 and exerts almost no influence on NCX3 (Iwamoto
*et al*, 2004b). In our structure, it binds to an allosteric binding site where most of the residues are conserved,
except for F248, which is substituted with L in NCX3 (Fig 4C). Functional experiments further confirm the crucial
role of this residue in subtype selectivity (Fig 4E), which provides a solid foundation for the rational design of
subtype-specific drugs. Previous investigations have demonstrated that the SEA0400 is preferable for inhibiting
the reverse mode of NCX (Tanaka, Nishimaru *et al.*, 2002), which is responsible for the uptake of extracellular
Ca^{2+} . However, the molecular basis for its selectivity towards the reverse mode over the forward mode remains
unknown. Understanding this molecular basis is crucial for rational development of pharmacological inhibitors that
specifically target NCX1 in the reverse mode, holding promising therapeutic potential in preventing Ca^{2+} overload
and electrical dysfunction in the context of ischemia/reperfusion injury and heart failure (Chen & Li, 2012a;
Imahashi *et al*, 2005). When the NCX operates in the reverse mode, the high concentration of intracellular Na^+
plays a dual role. On one hand, it is required to act as a driven ion, while on the other hand, it facilitates the
inactivation of NCX1.3 through the XIP region, thereby effectively preventing cells from Ca^{2+} overloading by NCX
during the pathological Na^+ accumulation (He *et al.*, 2000; Hilgemann *et al.*, 1992; Li *et al.*, 1991; Matsuoka *et al.*,
1993). As the sample is purified in the presence of high concentration Na^+ , we speculate that the complex structure
may represent an inactive state, primarily associated with the reverse mode. The XIP engages with the TM1-2
loop and plays a crucial role in the inactivation process by stabilizing the inward-facing conformation and
preventing a transition to the outward-facing state. Furthermore, these interactions between the XIP and TM1-2
loop also maintain a specific conformation of TM2ab, which is essential for the allosteric binding of SEA0400. Our
calcium imaging experiments have also demonstrated that the XIP is vital for the sensitivity of NCX to SEA0400
(Figs 4E and EV4F). In the forward mode, we speculate that the stability of the SEA0400 binding pocket is
compromised compared to the reverse mode, because the XIP may not effectively interact with the TM1-2 loop at
lower intracellular Na^+ , supported by the minimal inactivation observed in the forward mode over time (Iwamoto *et al.*,
2004a, Matsuoka, Nicoll *et al.*, 1995), thereby failing to stabilize the SEA0400 binding pocket. Hence, we
hypothesize that the Na^+ -dependent XIP-mediated inactivation, which is more prominently observed in the reverse
mode, leads to a more stable SEA0400 binding pocket compared to the forward mode, resulting in the selective
inhibition of the reverse mode by the SEA0400.”.

4. The main Figures have relatively small individual panels which makes it hard to see details described in the
manuscript when the paper is printed as hard copy. In particular, Figure 4 B showing electron density around the
small molecule SEA0400 is hard to see and it would make sense to either make it bigger, or alternatively add an
extra panel with an enlarged inset or add a larger Figure to the extended Figure S2 which shows density maps for
different protein regions, but not for the small molecule. In the current version, it is difficult to see how well the map
supports the proposed binding mode of SEA0400 and the mutant data mostly confirms the binding site but is
insufficient to confirm the exact pose of the molecule.

**Reply:** We appreciate reviewer’s comment. We have taken your suggestion and added a new panel in Appendix

Figure S2 (the original Figure S2) to show the map of the entire region surrounding the SEA0400 binding site from
two different views. The density around the SEA0400 binding site is well-defined and the surrounding amino acid
residues, including W210, F248, and V134, have clear side chain densities, confirming the reliability of the
SEA0400 binding. Furthermore, we have carefully examined the density of SEA0400 and found it to be consistent
with its structural features. Notably, the methoxyphenol group is connected to the ethoxyaniline group through an
oxygen atom, while the methoxy group serves as a linkage between the methoxyphenol group and the
difluorophenyl group. It should be noted that the flexibility of this linkage, particularly the methoxy group, could
potentially lead to weaker density at the connection site in the electron density map. This information is important
to consider when interpreting the density of SEA0400. However, we have thoroughly examined the density and
structural features of SEA0400 and have ensured that the overall binding pose remains consistent with the spatial
hindrance and interaction sites determined through functional experiments. The Appendix Figure S2E (the original
Figure S2E) is attached below for your convenience.

Figure 6*. The added Appendix Figure S2E. Density of residues and SEA0400 are shown as mesh.

**Minor points:**

1. Line 17: inferences -> infers

**Reply:** We thank reviewer's suggestion and have made the correction as "influences" in the line 17 of revised
manuscript.

2. Line 80: jaxtamembrane -> juxtamembrane

**Reply:** We thank the reviewer for pointing out this typo and we have corrected it in our revised manuscript.

3. Table S1: The clash score is a bit high - can this be improved by further optimization of the model in extra
refinement cycles?

**Reply:** Thank you for your comment. We have optimized the model by further refinement cycles and were able to
reduce the clash score from 14.61 to 12. We have updated the Table S1 accordingly.

4. Abstract, line 21 "stabilized at an inward-facing conformation" -> rephrase to "stabilized in an inward-facing
conformation"

**Reply:** We thank the reviewer for the comment and have corrected this typo.

5. Main Figure 2-4 title: structure basis -> structural basis

**Reply:** We thank the reviewer for the comment and have made the correction in our revised manuscript.

6. The authors may want to consider a more diverse choice of titles for these three main Figures.

**Reply:** We have provided a more diverse choice of titles for Figures 2-4. Figure 2 can be titled "Conformational
changes between the inward- and outward-facing of NCX1.3" or "Analysis of conformational transition of NCX1.3".
Figure 3 can be titled "Regulation of NCX1.3 by XIP region" or "Role of XIP in NCX1.3 regulation". Figure 4 can

be titled "Allosteric inhibition of NCX1.3 by SEA0400" or "SEA0400 binding pocket of the NCX1.3".

7. Figure 3 legend: descriptions for panels A-D in the legend do not match with the Figure which has panels A-E.
The description of panel B is missing (or part of A?) and the other panel labels need to be shifted to match the
Figure.

**Reply:** We thank the reviewer for pointing this out and have made a correction. At the time of submission, Panel
B was merged with Panel A. However, in the revised supplemental manuscript, we have corrected the labeling
and highlighted it to ensure clarity.

8. Figure 4 F: The color scheme is not ideal, because NCX1.3 is colored "in rainbow" and superposed with mjNCX
in cyan, but the rainbow looks more like a blue/pink gradient, making it hard to see the details in this relatively
small panel. The contrast is very faint too because of the semitransparent cartoon.

**Reply:** We thank the reviewer for this comment. We have revised Figure 4F and changed the color scheme of
NCX1.3 to a solid blue color for better contrast against the cyan mjNCX. Additionally, we have adjusted the
transparency of the cartoon representation to enhance visibility of the details in this panel. The revised Figure 4 is
attached below for your convenience.

9. Figure S2 B: maps created by cryo-EM are typically not called "electron" density maps, this is used for
crystallography, it is correct to call it instead "EM density maps" or "cryo-EM maps"

**Reply:** We thank reviewer's suggestion and have made the correction as "Local resolution map of NCX1.3^{SEA0400}."
in the revised manuscript.

10. Figure S4 Title: Sequence and structural alignment-> this figure does not show any sequence alignments, only
structural alignments.

**Reply:** We thank reviewer's comment. We have corrected this sentence in our revised manuscript.

Dear Yan,

Thank you for your patience during the re-review process of your manuscript (EMBOJ-2023-113785). We have now received the reports from the referees, which I copy below.

Despite the concerns of referee 1 over the purity of your sample and calcium ion binding, I would like to invite you to submit a second revision that addresses the concerns of referees 2 and 3 regarding the number of times the experiment has been repeated, and the binding pose of SEA0400. Please, though, also address the concerns of referee 1 in your point-by-point response.

Please contact me at any time during this second revision if you have further questions.

Best wishes,

William

William Teale, PhD
Editor
The EMBO Journal
w.teale@embojournal.org

We realize that it is difficult to revise to a specific deadline. In the interest of protecting the conceptual advance provided by the work, we recommend a revision within 3 months (30th Oct 2023). Please discuss the revision progress ahead of this time with the editor if you require more time to complete the revisions. Use the link below to submit your revision:

Referee #1:

Dong et al. EMBOJ-20230113785_r1

The revised manuscript suffers from the same fundamental problems of the previous version. While the text has been improved, the data have not.

This question from the first review has not been adequately addressed:

The presented data raise questions about the quality of the sample and how such issues may impact the structural interpretation. Fig. S1D shows two prominent bands that are not NCX, 70 kD and ~50 kD. What are these?

The authors have failed to provide a satisfactory explanation for the additional bands. It is rather disappointing that instead of providing data that would identify the bands, the authors simply wish them away as possible chaperones and minor contaminants. It is clear that they are at least 25% of the material, and while it is possible to remove 'junk' particles during processing, what the authors are actually doing is data selection. There are numerous anti-Chaperone antibodies that could be used to identify these bands, as well as straightforward mass spectrometry approaches. The explanation presented is not an acceptable level of rigor for the EMBO Journal.

This question from the first review has also not been adequately addressed:

The authors make claims about bound calcium ions but provide no evidence. Density to support the calcium ion assignment claims (line 140-145) is missing from the manuscript.

Despite the claims of a 'high affinity' binding sites (0.4 -10 μ M) and having Ca²⁺ present at a concentration that is many orders of magnitude above these binding affinities (1 mM), there is no experimental evidence for calcium binding. The inability to address this point directly leaves the authors with a low resolution structure, done in conditions that should have saturated the binding site many times over with calcium, but having no experimental evidence for calcium. Hence, the only argument they can make is by analogy to other, better defined, related structures. Without actual data to support their claim, any claims about calcium ions are simply speculative and not rigorously supported. It remains possible that despite the ordered conformation, calcium is simply not bound where they expect it to bind.

Referee #2:

The manuscript has been much improved by the authors following the reviewers comments, in particular there is a much more informative discussion.

I still note the following:

- 1) There is no reference to the figure showing the thermostability assay in the text. More importantly, there is no indication of repeats in the experiment. Has this just been done once?
- 2) While I am persuaded by the density and data for the SEA0400 molecule, there needs to be mention in the text regarding uncertainties in the exact binding mode, given the resolution and the breaks in the density.
- 3) The text related to the inverted repeats needs to reference the Forrest et al paper.

Referee #3:

The manuscript submitted by Dong et al. has improved substantially after an extensive revision, which has addressed most of the reviewer's comments. I still have a small concern regarding the suggested binding pose of SEA0400 which is not well supported by the EM density in this region according to the newly added enlarged Figure S2 E. The authors admit this and give an explanation, but I think it could also help to run a short MD simulation to confirm the stability of the pose they modelled into the map. This is good practice for putative compound-bound structures in drug discovery workflows and if the pose is indeed correct, it should remain stable during a 50 ns MD run. The interaction of SG0400 with the pocket in the transporter is likely a strong and stable one, given that the presence of the inhibitor increases the melting temperature of the purified protein significantly, and this should be reflected by stability in the MD run. The poor EM density for the inhibitor may likely also allow to

place the molecule upside down and still fit into the map equally well, but it will probably be unstable in MD if it is incorrect and fails to engage into stabilizing interactions with the surrounding residues. For further improvement, I also suggest that the authors add a concluding sentence summarizing the major insights and conclusions from this study at the end of the discussion which is now expanded and covers most of the suggested topics, but it ends very abruptly in its current version. Once these issues are addressed, the manuscript is suitable for publication.

Point-by-point response for

Mechanistic insight into allosteric inhibition of human sodium-calcium exchanger NCX1 by XIP and SEA0400

**Referee #1:**

Dong et al. EMBOJ-20230113785_r1

The revised manuscript suffers from the same fundamental problems of the previous version. While the text has
been improved, the data have not.

**Reply:** We thank the reviewer for recognizing the improvements in our manuscript and for his/her comments to
further enhance it.

This question from the first review has not been adequately addressed:

The presented data raise questions about the quality of the sample and how such issues may impact the structural
interpretation. Fig. S1D shows two prominent bands that are not NCX, 70 kD and ~50 kD. What are these? The
authors have failed to provide a satisfactory explanation for the additional bands. It is rather disappointing that
instead of providing data that would identify the bands, the authors simply wish them away as possible chaperones
and minor contaminants. It is clear that they are at least 25% of the material, and while it is possible to remove
'junk' particles during processing, what the authors are actually doing is data selection. There are numerous anti-
Chaperone antibodies that could be used to identify these bands, as well as straightforward mass spectrometry
approaches. The explanation presented is not an acceptable level of rigor for the EMBO Journal.

**Reply:** We appreciate the reviewer's comment. In this revised manuscript, we have employed mass spectrometry
approaches to identify these protein bands, as suggested by the reviewer. To achieve this, we purified the NCX1.3
protein. The protein band that appears on the SDS-PAGE gel around 100 kD is consistent with NCX1.3. Moreover,
there are several distinct, closely spaced bands around the 70 kD and 50 kD marker bands (Figure 1* A). We
excised a broader region (shown in Figure 1* A) for the mass spectrometry analysis. The results suggest that the
proteins with a lower molecular weight represent degraded NCX1.3, heat shock protein 70 (HSP70), and a few
other contaminant proteins (Figure 1* B).

Figure 1*. The results of Coomassie-blue-stained SDS-PAGE gel for three separate batches of the NCX 1.3 and
mass spectrometry analysis. A. Coomassie-blue-stained SDS-PAGE gel for mass spectrometry analysis. B. The
results of mass spectrometry analysis for NCX1.3, which only displayed a PSM (Peptide-Spectrum Match) score
greater than 10.

In the revised manuscript, we have included this result and added a brief discussion. Please see Pg 4 (lines 101-
106), it now reads "Size-exclusion chromatography (SEC) displayed monodisperse peaks, and SDS-PAGE result
confirmed the intactness of the full-length NCX1.3 (Appendix Fig S1C and D). However, there are several distinct,
closely spaced bands around the 70 kD and 50 kD marker bands. To identify these protein bands, we excised a

broader region (Appendix Fig S1D) for the mass spectrometry analysis. The results suggest that the proteins with a lower molecular weight represent degraded NCX1.3, heat shock protein 70 (HSP70), and a few other contaminant proteins (Appendix Fig S1E).".

In the revised manuscript, we also incorporated the SDS-PAGE gel used for the mass spectrometry assay as Appendix Fig S1D and have incorporated the mass spectrometry results into this figure (Appendix Fig S1E). The revised Appendix Figure S1 is attached here for your convenience.

Figure 2*. Revised versions of Appendix Figure S1.

- A** Representative Na⁺ dependence of outward Na⁺-Ca²⁺ exchange current traces in HEK293T cells transfected with NCX1.3^{WT} and untransfected as negative control by whole-cell patch clamp. Outward currents were activated by changing the extracellular Ca²⁺ from 0 to 1 mM for 10s.
- B** Representative traces of cytosolic Ca²⁺ measurements using Fura-2 AM as an indicator in HEK293T cells transfected with NCX1.3^{WT} and untransfected as negative control. The change of solution was marked by time breaks. The Krebs' buffer including 160 mM NaCl was changed Na⁺-free NMDG⁺ buffer (0 mM NaCl) at one minute. Subsequently, 500 μM ATP was added at the end as a control for cell viability. Data represent the mean ± SEM of recordings. The Fura2 ratio (340/380) was used as a quantitative indicator of intracellular [Ca²⁺].
- C** Size-exclusion chromatogram (Superose 6 increase) of the purified protein sample of NCX1.3. The peak 1 (marked within black dashed lines) was pooled and concentrated for cryo-EM study. The peak 2 represents cleaved mCherry.
- D** Coomassie-blue-stained SDS-PAGE gel of the NCX 1.3. The protein band is labeled. The experiments were repeated independently more than 3 times with similar results.
- E** The results of mass spectrometry analysis for NCX1.3, which only displayed a PSM (Peptide-Spectrum Match) score greater than 10.
- F** Thermal stability assay as described in the methods section. The addition of 100 μM SEA0400 increased the thermal stability of NCX1.3. The experiments were repeated independently 4 times, consistently yielding similar results.

This question from the first review has also not been adequately addressed:

The authors make claims about bound calcium ions but provide no evidence. Density to support the calcium ion
assignment claims (line 140-145) is missing from the manuscript. Despite the claims of a 'high affinity' binding
sites (0.4 -10 μM) and having Ca^{2+} present at a concentration that is many orders of magnitude above these
binding affinities (1 mM), there is no experimental evidence for calcium binding. The inability to address this point
directly leaves the authors with a low resolution structure, done in conditions that should have saturated the binding
site many times over with calcium, but having no experimental evidence for calcium. Hence, the only argument
they can make is by analogy to other, better defined, related structures. Without actual data to support their claim,
any claims about calcium ions are simply speculative and not rigorously supported. It remains possible that despite
the ordered conformation, calcium is simply not bound where they expect it to bind.

**Reply:** We appreciate reviewer's comment. While structural analysis supports the presence of Ca^{2+} in NCX1.3
structure, we totally agree with the reviewer that it's possible the calcium ions may not bind or may do so differently
than we proposed. Consequently, we have updated our discussion in the revised manuscript.

In the revised manuscript, we adjusted our related discussion. Please see Pg 5 (lines 145-159), it now reads "The
CBD1 and CBD2 are organized in a head-to-tail tandem through a short linker, generating a $\sim 126^\circ$ angle between
CBD1 and CBD2 domains (Figures 1A and EV1). The CBD1 domain of the NCX1 protein contains four calcium
ion binding sites, with Ca^{2+} ions displaying a strong affinity for CBD1 ($K_{d_{\text{Ca}1/\text{Ca}2}}$: $\sim 10 \mu\text{M}$, $K_{d_{\text{Ca}3/\text{Ca}4}}$: $\sim 0.4 \mu\text{M}$) (Hilge
*et al.*, 2009a). However, due to the moderate resolution of our resolved NCX1.3 map, we were unable to observe
the density corresponding to Ca^{2+} ions. A previous study suggested that the upper half of isolated CBD1 loses its
structural integrity in the absence of Ca^{2+} (Hilge *et al.*, 2006). In contrast, the CBD1 domain in our NCX1.3 structure
exhibits a well-defined conformation, with an RMSD of approximately 0.9 Å for 116 C α -pairs when compared to
the isolated Ca^{2+} -bound CBD1 (Figure EV1A). Considering these findings, along with the fact that we added
approximately 1 mM Ca^{2+} during the sample preparation, we speculate that Ca^{2+} ions likely bind to the CBD1
domain in our elucidated NCX1.3 structure. Nevertheless, we cannot entirely rule out the possibility that Ca^{2+} may
remain unbound or be located in alternative positions. Further investigation is required to understand calcium
binding site and its regulatory roles."

**Referee #2:**

The manuscript has been much improved by the authors following the reviewers comments, in particular there is
a much more informative discussion.

I still note the following:

1) There is no reference to the figure showing the thermostability assay in the text. More importantly, there is no
indication of repeats in the experiment. Has this just been done once?

**Reply:** We thank the reviewer for pointing this out. In the revised manuscript, we have incorporated a reference
to this figure. Please see Pg 4 (lines 106-108), it now reads "Meanwhile, the thermofluor assay demonstrated that
the NCX1.3-selective inhibitor SEA0400 can raise melting temperature of NCX1.3 by $\sim 10^\circ\text{C}$ (Appendix Fig S1F)".
Regarding the repetition of the thermal stability assay, we actually carried out this the experiment four times using
the same method as described in the methods section (Figure 3*). All four of these experiments yielded similar
results, indicating that addition of 100 μM SEA0400 raised the melting temperature of NCX1.3 by approximately
10 $^\circ\text{C}$ (Figure 3* A-D). In the manuscript, only one representative result was included (Figure 3* A).

To clarify repetition of the thermal stability assay, we revised the figure legend. It now reads (Lines 42-43) "Thermal

stability assay as described in the methods section. The addition of 100 μ M SEA0400 increased the thermal
 stability of NCX1.3. The experiments were repeated independently 4 times, consistently yielding similar results.”.

 Figure 3*. Thermal stability assay of four repeats as described in the methods section. The addition of 100 μ M
 SEA0400 increased the thermal stability of NCX1.3. A. The representative profile of thermal stability assay
 included in the manuscript. B-D. Thermal stability assay of the other three repeats.

2) While I am persuaded by the density and data for the SEA0400 molecule, there needs to be mention in the text
 regarding uncertainties in the exact binding mode, given the resolution and the breaks in the density.

**Reply:** We thank the reviewer for pointing this out and agree that it's important to acknowledge potential
 uncertainties in the binding mode of SEA0400 in the manuscript. According to Reviewer3's suggestion, we
 conducted 100 ns molecular dynamics simulations for our structure as well as a model with inverted binding modes
 of SEA0400. The MD simulation results support the reliability of our SEA0400 modeling (Figure 4* A-D). We
 prepared a figure about MD analysis and included it as Appendix Fig S4 in the revised manuscript. We have also
 attached here for your review (Figure 4*).

**Figure 4*. Molecular dynamic simulation analysis of SEA0400 binding to NCX1.3.**

A Initial structures used in MD simulations. Two sets of simulations, referred to as the NCX1.3^{EM} and NCX1.3^{rev},
 were included in the MD study with SEA0400 initially placed in the pocket in two distinct orientations. For
 NCX1.3^{EM}, SEA0400 adopts the same poses as observed in the cryo-EM structure (SEA0400^{EM}). On the

other hand, in the NCX1.3^{rev}, SEA0400 assumed a reversed pose (SEA0400^{rev}).
B R.M.S.D. plots for ligand and protein backbone during MD simulation. R.M.S.D. plots were generated for both
SEA0400 and the protein backbone. The calculation of backbone R.M.S.D. for NCX1.3 using residues 51-
283, 683-720, and 736-937, with the initial structure serving as the reference. The calculation of SEA0400
R.M.S.D. involved the heavy atoms of SEA0400, using its initial structure as the reference. Results from three
independent trajectories were distinguished by different colors.
C Cluster populations obtained from clustering analysis. The clustering analysis was conducted respectively for
NCX1.3^{EM} and NCX1.3^{rev}, each involving three independent 100 ns trajectories. Heavy atoms from SEA0400
and the pocket (protein residues within 5 Å of SEA0400) were employed in the clustering analysis, with a 1.5
Å R.M.S.D. cut-off.
D Structural comparisons of initial poses of SEA0400 with representative poses derived from clustering analysis.
The left panel demonstrates the high consistency between the cryo-EM structure and the representative
structure from the NCX1.3^{EM}, indicating the high stability of SEA0400 binding in the observed cryo-EM pose.
Conversely, the right panel depicts diverse poses observed in the NCX1.3^{rev}, suggesting the low stability of
such a binding pose. The NCX1.3 model is shown in cartoon and SEA0400 is shown in sticks.

In the revised manuscript, we have incorporated a brief discussion of MD simulations. Please see Pgs 8-9 (lines
253-269), it now reads “Considering the elongated, strip-shaped structure of SEA0400 and moderate map quality
for this molecule, we carried out molecular dynamics simulations to validate its binding pose. These simulations
were initially performed using the cryo-EM structure of SEA0400-bound NCX1.3 (NCX1.3^{EM}). Additionally, we also
studied an alternative model in which SEA0400 binds to the same pocket but adopts an inverted orientation
compared to what was observed in the cryo-EM structure (NCX1.3^{rev}) (Appendix Fig S4A). Three independent
trajectories, each lasting 100 ns, were conducted for both NCX1.3^{EM} and NCX1.3^{rev}, respectively. Subsequently,
the trajectories for each group were combined and subjected to a clustering analysis (Appendix Fig S4B and C).
The analysis focused on the ligand and protein residues within a 5 Å radius of the ligand, using a 1.5 Å R.M.S.D.
cut-off. The trajectory clustering for NCX1.3^{EM} yielded only two clusters, with cluster 1 being the predominant
conformation. This conformation is close to the pose of SEA0400 observed in our cryo-EM structure, indicating
the high stability of SEA0400 binding in the observed cryo-EM pose (Appendix Fig S4C and D). In contrast, the
trajectory for NCX1.3^{rev} produced eight distinct clusters, and the pose of SEA0400 exhibited significant deviations
among these clusters, suggesting the low stability of such a binding pose (Appendix Fig S4C and D). These
findings provide support for the binding pose of SEA0400 in the cryo-EM structure.”.

We have also included a method section for the MD simulations. Please see Pg 18 (lines 557-590), it now reads
“We created an alternate model by reversing the orientation of SEA0400 (NCX1.3^{rev}), specifically flipping it upside
down compared to its original positioning in the cryo-EM structure. In this model, the difluorophenyl group was
inserted deeply into the binding pocket, while the 3-ethosyanilion group was directed toward the intracellular side.
The cryo-EM structure of SEA0400-bound NCX1.3 (NCX1.3^{EM}) and model of NCX1.3^{rev} were used as the initial
structure for molecular dynamics simulation. In modeling process, due to the intracellular structural domains CBD1
and CBD2 were not implicated in SEA0400 binding, we removed them, and retained the residues 51-283, 683-
720, and 736-937 for subsequent analysis. The molecular dynamics simulations were conducted using
GROMACS v.2021.6(Abraham, Murtola *et al.*, 2015) with the Amber ff14SB force field(Maier, Martinez *et al.*, 2015)
for protein and lipids and GAFF2(Wang, Wang *et al.*, 2006) for SEA0400. All the input files were generated by
CHARMM-GUI(Jo, Kim *et al.*, 2008). The cryo-EM structure NCX1.3^{EM} and model of NCX1.3^{rev} were separately

embedded into a pre-equilibrated 1-palmitoyl-2-oleoyl-sn-glycero-3-phosphatidylcholine membrane (POPC) with
the membrane orientation calculated by the Orientations of Proteins in Membranes database (PPM 2.0) (Lomize,
Pogozeva *et al.*, 2012). All systems were minimized for 5,000 steps. Equilibrations with weak restraints were
conducted before running production MD, following standard CHARMM-GUI membrane equilibration steps. The
configuration of the simulation box was chosen as rectangular. The system was then solvated with TIP3P
water(Jorgensen, Chandrasekhar *et al.*, 1983), and 150 mM NaCl was incorporated to neutralize the system using
the Monte Carlo method. In the final NCX1.3^{EM} system, there were a total of 62,973 atoms, including 113 POPCs,
13418 water molecules, 73 counter ions, the transmembrane region of NCX1.3^{EM}, and SEA0400^{EM}. The final
NCX1.3^{rev} system contained 62,756 atoms in total, which consisted of 114 POPCs, 13,301 water molecules, 73
counter ions, the transmembrane region of NCX1.3^{rev}, and SEA0400^{rev}. In each system, three separate 100 ns
unrestrained production simulations were conducted, maintaining a constant particle number, 1 bar pressure, and
303.15 K temperature. The velocity-rescaling thermostat(Bussi, Donadio *et al.*, 2007) and Parrinello-Rahman
barostat(Parrinello & Rahman, 1980) were employed for temperature and pressure coupling, respectively. Long-
range electrostatic interactions were calculated using the particle mesh Ewald (PME) method. A cut-off of 12 Å
was established for van der Waals interactions, incorporating a force-switching function at 10 Å. The LINES
algorithm (Hess, 2008) was utilized to keep bonds involving hydrogen fixed during each 2 fs integration time step.
Clustering analysis was carried out using the GROMOS software(Daura, Gademann *et al.*, 1999).”.

3) The text related to the inverted repeats needs to reference the Forrest et al paper.

**Reply:** We thank the reviewer for pointing this out and have made the correction. In the revised manuscript, we
have added the corresponding reference, please see Pg 4 (line 174), it now reads “We created an outward-facing
model of NCX1.3 (NCX1.3^{OF*}) by superimposing TM2-TM5 on TM7-TM10, and vice versa (Liao *et al.*, 2012;
Forrest *et al.*, 2008; Hu *et al.*, 2011) (Fig EV2A).”.

**Referee #3:**

The manuscript submitted by Dong et al. has improved substantially after an extensive revision, which has
addressed most of the reviewer's comments. I still have a small concern regarding the suggested binding pose of
SEA0400 which is not well supported by the EM density in this region according to the newly added enlarged
Figure S2E. The authors admit this and give an explanation, but I think it could also help to run a short MD
simulation to confirm the stability of the pose they modelled into the map. This is good practice for putative
compound-bound structures in drug discovery workflows and if the pose is indeed correct, it should remain stable
during a 50 ns MD run. The interaction of SG0400 with the pocket in the transporter is likely a strong and stable
one, given that the presence of the inhibitor increases the melting temperature of the purified protein significantly,
and this should be reflected by stability in the MD run. The poor EM density for the inhibitor may likely also allow
to place the molecule upside down and still fit into the map equally well, but it will probably be unstable in MD if it
is incorrect and fails to engage into stabilizing interactions with the surrounding residues. For further improvement,
I also suggest that the authors add a concluding sentence summarizing the major insights and conclusions from
this study at the end of the discussion which is now expanded and covers most of the suggested topics, but it ends
very abruptly in its current version. Once these issues are addressed, the manuscript is suitable for publication.

**Reply:** We greatly appreciate the reviewer’s comment and his/her valuable suggestions for improving our
manuscript.

We totally agree with reviewer that the MD simulation would be valuable in verifying the binding pose of SEA0400.
Therefore, we created a NCX1.3 model with inverted orientation of SEA0400 (NCX1.3^{rev}), in comparison to our

cryo-EM structure (Figure 4* A). Both the cryo-EM structure of SEA0400 bound NCX1.3 (NCX1.3^{EM}) and the
 NCX1.3^{rev} model are subjected to molecular simulations analysis. Subsequently, we embedded two distinct
 structural models, each with different ligand orientations, into membrane systems and conducted three individual
 100-nanosecond molecular dynamics simulations for each model. Clustering analysis of these trajectories
 demonstrated that binding mode of SEA0400 in current NCX1.3^{EM} structure was stable during the dynamic process,
 while it was not stable in NCX1.3^{rev} model (Figure 4* B-D). These findings strongly support the reliable modeling
 of SEA0400 in our structure.

In the revised manuscript, we have included a brief discussion about molecular simulations. Please see Pgs 8-9
 (lines 253-269), it now reads “Considering the elongated, strip-shaped structure of SEA0400 and moderate map
 quality for this molecule, we carried out molecular dynamics simulations to validate its binding pose. These
 simulations were initially performed using the cryo-EM structure of SEA0400-bound NCX1.3 (NCX1.3^{EM}).
 Additionally, we also studied an alternative model in which SEA0400 binds to the same pocket but adopts an
 inverted orientation compared to what was observed in the cryo-EM structure (NCX1.3^{rev}) (Appendix Fig S4A).
 Three independent trajectories, each lasting 100 ns, were conducted for both NCX1.3^{EM} and NCX1.3^{rev},
 respectively. Subsequently, the trajectories for each group were combined and subjected to a clustering analysis
 (Appendix Fig S4B and C). The analysis focused on the ligand and protein residues within a 5 Å radius of the
 ligand, using a 1.5 Å R.M.S.D. cut-off. The trajectory clustering for NCX1.3^{EM} yielded only two clusters, with cluster
 1 being the predominant conformation. This conformation is close to the pose of SEA0400 observed in our cryo-
 EM structure, indicating the high stability of SEA0400 binding in the observed cryo-EM pose (Appendix Fig S4C
 and D). In contrast, the trajectory for NCX1.3^{rev} produced eight distinct clusters, and the pose of SEA0400 exhibited
 significant deviations among these clusters, suggesting the low stability of such a binding pose (Appendix Fig S4C
 and D). These findings provide support for the binding pose of SEA0400 in the cryo-EM structure.”.

We prepared a figure about MD analysis and included it as Appendix Fig S4 in the revised manuscript. We have
 also attached here for your review.

Figure 4*. Molecular dynamic simulation analysis of SEA0400 binding to NCX1.3.

- A Initial structures used in MD simulations. Two sets of simulations, referred to as the NCX1.3^{EM} and NCX1.3^{rev},
were included in the MD study with SEA0400 initially placed in the pocket in two distinct orientations. For
NCX1.3^{EM}, SEA0400 adopts the same poses as observed in the cryo-EM structure (SEA0400^{EM}). On the
other hand, in the NCX1.3^{rev}, SEA0400 assumed a reversed pose (SEA0400^{rev}).
- B R.M.S.D. plots for ligand and protein backbone during MD simulation. R.M.S.D. plots were generated for both
SEA0400 and the protein backbone. The calculation of backbone R.M.S.D. for NCX1.3 using residues 51-
283, 683-720, and 736-937, with the initial structure serving as the reference. The calculation of SEA0400
R.M.S.D. involved the heavy atoms of SEA0400, using its initial structure as the reference. Results from three
independent trajectories were distinguished by different colors.
- C Cluster populations obtained from clustering analysis. The clustering analysis was conducted respectively for
NCX1.3^{EM} and NCX1.3^{rev}, each involving three independent 100 ns trajectories. Heavy atoms from SEA0400
and the pocket (protein residues within 5 Å of SEA0400) were employed in the clustering analysis, with a 1.5
Å R.M.S.D. cut-off.
- D Structural comparisons of initial poses of SEA0400 with representative poses derived from clustering analysis.
The left panel demonstrates the high consistency between the cryo-EM structure and the representative
structure from the NCX1.3^{EM}, indicating the high stability of SEA0400 binding in the observed cryo-EM pose.
Conversely, the right panel depicts diverse poses observed in the NCX1.3^{rev}, suggesting the low stability of
such a binding pose. The NCX1.3 model is shown in cartoon and SEA0400 is shown in sticks.

In the revised manuscript, we've included a MD simulations method section. Please see Pg 18 (lines 557-590), it
now reads, "We created an alternate model by reversing the orientation of SEA0400 (NCX1.3^{rev}), specifically
flipping it upside down compared to its original positioning in the cryo-EM structure. In this model, the
difluorophenyl group was inserted deeply into the binding pocket, while the 3-ethosyanilion group was directed
toward the intracellular side. The cryo-EM structure of SEA0400-bound NCX1.3 (NCX1.3^{EM}) and model of
NCX1.3^{rev} were used as the initial structure for molecular dynamics simulation. In modeling process, due to the
intracellular structural domains CBD1 and CBD2 were not implicated in SEA0400 binding, we removed them, and
retained the residues 51-283, 683-720, and 736-937 for subsequent analysis. The molecular dynamics simulations
were conducted using GROMACS v.2021.6(Abraham, Murtola *et al.*, 2015) with the Amber ff14SB force field(Maier,
Martinez *et al.*, 2015) for protein and lipids and GAFF2(Wang, Wang *et al.*, 2006) for SEA0400. All the input files
were generated by CHARMM-GUI(Jo, Kim *et al.*, 2008). The cryo-EM structure NCX1.3^{EM} and model of NCX1.3^{rev}
were separately embedded into a pre-equilibrated 1-palmitoyl-2-oleoyl-sn-glycero-3-phosphatidylcholine
membrane (POPC) with the membrane orientation calculated by the Orientations of Proteins in Membranes
database (PPM 2.0) (Lomize, Pogozheva *et al.*, 2012). All systems were minimized for 5,000 steps. Equilibrations
with weak restraints were conducted before running production MD, following standard CHARMM-GUI membrane
equilibration steps. The configuration of the simulation box was chosen as rectangular. The system was then
solvated with TIP3P water(Jorgensen, Chandrasekhar *et al.*, 1983), and 150 mM NaCl was incorporated to
neutralize the system using the Monte Carlo method. In the final NCX1.3^{EM} system, there were a total of 62,973
atoms, including 113 POPCs, 13418 water molecules, 73 counter ions, the transmembrane region of NCX1.3^{EM},
and SEA0400^{EM}. The final NCX1.3^{rev} system contained 62,756 atoms in total, which consisted of 114 POPCs,
13,301 water molecules, 73 counter ions, the transmembrane region of NCX1.3^{rev}, and SEA0400^{rev}. In each
system, three separate 100 ns unrestrained production simulations were conducted, maintaining a constant
particle number, 1 bar pressure, and 303.15 K temperature. The velocity-rescaling thermostat(Bussi, Donadio *et al.*,
2007) and Parrinello-Rahman barostat(Parrinello & Rahman, 1980) were employed for temperature and

pressure coupling, respectively. Long-range electrostatic interactions were calculated using the particle mesh
Ewald (PME) method. A cut-off of 12 Å was established for van der Waals interactions, incorporating a force-
switching function at 10 Å. The LINES algorithm (Hess, 2008) was utilized to keep bonds involving hydrogen fixed
during each 2 fs integration time step. Clustering analysis was carried out using the GROMOS software(Daura,
Gademann *et al.*, 1999).”.

We agree with review that it would be good to add a concluding sentence. In the revised manuscript, we have
added a sentence to summarize our findings, please see Pg 12 (lines 396-400), it now reads “In summary, our
research demonstrates that in the reverse mode, the high concentration of intracellular Na⁺ promotes XIP
interactions with TMD and CBD2, leading to inactivation of NCX1.3 and stabilization of the allosteric binding pocket
of SEA0400, thereby elucidating the molecular basis of the synergistic inhibition of NCX1.3 by XIP and SEA0400,
with SEA0400 selectively targeting NCX1.3 in the reverse mode.”.

Dear Prof. Zhao,

We have now received re-review reports from the three referees that initially appraised your manuscript. As you will see, you have addressed their concerns satisfactorily, although minor changes to your discussion are recommended. Before I can finally accept the manuscript though, there are some small editorial points that remain and need to be addressed. In this regard would you please:

- add up to five keywords,
- change the 'conflict of interest statement' to the 'disclosure and competing interests statement',
- remove the author credit section from the manuscript,
- include figure callouts for Appendix Table S1-S2 in the main text,
- reorganize Source Data files to one file/folder per figure and ZIP for each main figure. For EV and/or appendix figures, ZIP together all source data,
- provide a synopsis image (550 pixels wide by 200-400 high),
- provide a two-sentence general summary statement and 3-5 bullet points that capture the key findings of the paper, and
- rename movie file to Movie EV1 with the corresponding callout; ZIP legend with the movie file.

Best wishes,

William Teale

William Teale, PhD
Editor
The EMBO Journal
w.teale@embojournal.org

We realize that it is difficult to revise to a specific deadline. In the interest of protecting the conceptual advance provided by the work, we recommend a revision within 3 months (1st Feb 2024). Please discuss the revision progress ahead of this time with the editor if you require more time to complete the revisions. Use the link below to submit your revision:

Referee #1:

The issues from the prior rounds are addressed. I have no further comments.

Referee #2:

My previous comments have been adequately addressed. The MD study complements the work well.

I actually found the new summarising sentence at the end difficult to read. The authors may like to modify this slightly.

Referee #3:

The remaining issues after the first revision have now been addressed and the manuscript can be accepted. I suggest that the authors cite a recent related study of NCX1.1 and discuss similarities and differences to the structure of NXC1 reported in this independent study: <https://doi.org/10.1038/s41467-023-41885-4>

Especially the section in the discussion speculating about conformational changes in the CBDs in response to Ca²⁺ for different splice isoforms of NCX would benefit from this comparison since the other publication reports structures of human cardiac NCX1.1 in both inactivated (apo) and activated (Ca bound CBD) states.

Point-by-point response for

Mechanistic insight into allosteric inhibition of human sodium-calcium exchanger NCX1 by XIP and SEA0400

Referee #1:

The issues from the prior rounds are addressed. I have no further comments.

Reply: We greatly appreciate the reviewer's positive comment.

Referee #2:

My previous comments have been adequately addressed. The MD study complements the work well.

Reply: We are thankful to the reviewer for his/her kind and encouraging comments.

I actually found the new summarising sentence at the end difficult to read. The authors may like to modify this slightly.

Reply: We appreciate the reviewer's comment and pointing this out. In the revised manuscript, we have modified the concluding sentence, please see Pg 12 (lines 395-399), it now reads "In summary, our research demonstrates that in the reverse mode, the high concentration of intracellular Na⁺ promotes XIP interactions with TMD and CBD2, leading to inactivation of NCX1.3. This process also stabilizes the allosteric binding pocket of SEA0400, underlying the molecular basis for the synergistic inhibition of NCX1.3 by XIP and SEA0400, with SEA0400 selectively targeting NCX1.3 in the reverse mode."

Referee #3:

The remaining issues after the first revision have now been addressed and the manuscript can be accepted. I suggest that the authors cite a recent related study of NCX1.1 and discuss similarities and differences to the structure of NXC1 reported in this independent study: <https://doi.org/10.1038/s41467-023-41885-4>

Especially the section in the discussion speculating about conformational changes in the CBDs in response to Ca²⁺ for different splice isoforms of NCX would benefit from this comparison since the other publication reports structures of human cardiac NCX1.1 in both inactivated (apo) and activated (Ca bound CBD) states.

Reply: We sincerely appreciate the reviewer's comment and his/her valuable suggestions for improving our manuscript. In the revised manuscript, we have included a discussion comparing the conformational changes in the CBDs for NCX1.1 and NCX1.3 (Figure 1*), please see Pgs 6-7 (lines 157-166), it now reads "By superimposing the invariant CBD1 domains of NCX1.3, NCX1.4, and NCX1.1^[1] (its structure was available online during our review), the angle between CBD1 and CBD2 decreases by 7° in NCX1.4 (exon AD), but

increase by 10° in NCX1.1 (exon ACDEF), which may be attributed to the previously reported splicing region within CBD2 (Kofuji *et al*, 1994; Quednau *et al.*, 1997; Xue *et al*, 2023) (Figs EV1A–D). The CBD2 connects to TM6 via three helices (H1–H3), forming a ~37° angle between CBD2 and the membrane plane (Fig 1A), which is larger than that observed in NCX1.1, possibly due to different local interactions between the TMD region and mutually exclusive exons A (NCX1.1) and B (NCX1.3), or the truncation of NCX1.1 between CBD1 and TMD (Fig EV1D).".

[1] Xue J, Zeng W, Han Y, John S, Ottolia M, Jiang Y. Structural mechanisms of the human cardiac sodium-calcium exchanger NCX1. *Nat Commun.* 2023 Oct 4;14(1):6181.

In the revised manuscript, we prepared a figure about structural comparison of CBD domains of NCX1.1 and NCX1.3 and included it as Figure EV1 in the revised manuscript. We have also attached here for your convenience.

Figure 1*. Structural alignment of CBDs among human NCX1 isoforms. Related to Figure 1.

- A Structural comparison of CBDs between NCX1.3 and NCX1. The Ca²⁺-bound CBD1 structure of NCX1 (PDB ID: 2DPK) is colored in yellow and the bound Ca²⁺ ions are displayed as yellow spheres. CBD1 and CBD2 of NCX1.3 in our structure are colored in green and dark green, respectively. The CBD2 of NCX1.3 (PDB ID: 2KLT) is colored in pink.
- B Structural comparison of CBDs between NCX1.3 and NCX1.4 (CBD12^{E454K}, PDB ID:

3US9) using the invariant CBD1 domains. CBD1 and CBD2 of NCX1.4 are colored in purple. The key residues are shown as sticks. The bound Ca^{2+} ions of CBD1 for NCX1.4^{E454K} are displayed as purple spheres. The angles between CBD1 and CBD2 of NCX1.3 and NCX1.4 are marked, respectively.

- C Structural comparison of CBDs between NCX1.3 and NCX1.1 (PDB ID: 8SGJ). CBD1 and CBD2 of NCX1.3 in our structure are colored in green and dark green, respectively.
- D Structure superposition of NCX1.3 and NCX1.1 (PDB ID: 8SGJ) using TMD domain. NCX1.3 and NCX1.1 are shown as salmon and grey cartoon, respectively.

Dear Yan,

I am pleased to inform you that your manuscript has been accepted for publication in the EMBO Journal.

Congratulations on your beautiful work!

Best wishes,

William

William Teale, PhD
Editor
The EMBO Journal
w.teale@embojournal.org
